



# Climate Bifurcations in a Schwarzschild Equation Model of the Arctic Atmosphere

Kolja L. Kypke[1], William F. Langford[2], Gregory M. Lewis[3], and Allan R. Willms[2]

[1]Niels Bohr Institute, University of Copenhagen, Copenhagen, Denmark
[2]Dept. of Mathematics & Statistics, University of Guelph, Guelph, Canada
[3]Faculty of Science, Ontario Tech University, Oshawa, Canada

**Correspondence:** Allan R. Willms (AWillms@uoguelph.ca)

**Abstract.** A column model of the Arctic atmosphere-ocean system is developed including the nonlinear responses of surface albedo and water vapor to temperature. The atmosphere is treated as a gray gas and the flux of longwave radiation is governed by the two-stream Schwarzschild equations. Representative carbon pathways (RCPs) are used to model carbon dioxide concentrations into the future. The resulting nine-dimensional two-point boundary value problem is solved under various RCPs and the solutions analyzed. The model predicts that under the highest carbon pathway, the Arctic climate will undergo an irreversible bifurcation to a warm steady state, which would correspond to an annually ice-free situation. Under the lowest carbon pathway, corresponding to very aggressive carbon emission reductions, the model exhibits only a mild increase in Arctic temperatures. Under the two moderate carbon pathways, temperatures increase more substantially, and the system enters a region of bistability where external perturbations could possibly cause an irreversible switch to a warm, ice-free state.

## 1 Introduction

Climate change is causing rapid temperature increases in the polar regions. A fundamental question is whether these temperature increases are reversible. If humanity fails to prevent a substantial warming of the planet in the next few decades, which is appearing to be more and more likely, will it be possible in the future to reverse our effects on climate enough to restore lower temperatures? Or will we have passed a tipping point beyond which return to the present state is impossible? We address this question in particular for the Arctic, where the observed climate change is the most dramatic.

The Earth's climate is an extremely complex system. Modelling efforts range from simple models attempting to isolate the most pertinent features, to very complicated numerical models trying to capture as many details as possible. The model presented here is close to the simple end of this spectrum, although not as simple as some, in that it is a nine-dimensional nonlinear two-point boundary value problem. The advantage of relatively simple models is that they allow more direct analysis of cause and effect, which is often obscured in highly complicated models.

The term "tipping point" is used by different researchers in various ways; see Russill (2015) and Lenton et al. (2008) for some definitions and discussion of the term. In all cases however, tipping points are associated with large qualitative changes in a system due to relatively small changes in the parameters, or "forcings" that drive the system. In the present paper, tipping points arise as a result of saddle-node and cusp bifurcations in the mathematical model. The mathematical theory of




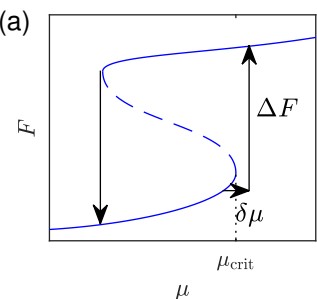
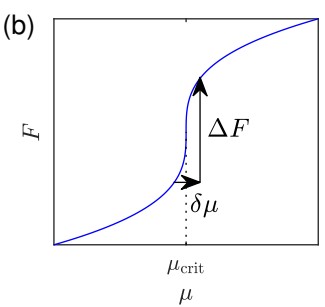
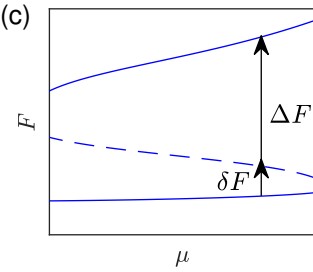

**Figure 1.** (a) Hysteresis arising from two saddle-node bifurcations, (b) cusp bifurcation, (c) bistable system. Solid curves indicate stable states, and dashed curves are unstable states. In (a) and (b) a small change $\delta\mu$ in the control parameter near the bifurcation value $\mu_{\mathrm{crit}}$ causes a large change $\Delta F$ in the system. In (a) the return bifurcation happens at a different value of $\mu$. In (c) a small perturbation $\delta F$ in the system causes a large change $\Delta F$.

bifurcations is well-developed (Kuznetsov, 2004) and employed here. Figure 1 illustrates the typical behavior associated with these bifurcations. In Fig. 1(a) there are two saddle-node bifurcations resulting in a parameter interval of bistability, that is, two stable states coexist for an interval of $\mu$ values. If the system is on the lower stable state, then, as $\mu$ increases through a critical value $\mu_{\mathrm{crit}}$, there is an abrupt jump to the upper stable state. In contrast, as $\mu$ decreases, the jump back to the lower state does not occur until a much smaller critical value of $\mu$. This phenomenon is called hysteresis. If present in the Earth's

climate system, it implies that once the upward jump occurs, it may be very difficult to achieve the reverse jump back to the original climate state. Fig. 1(b) illustrates the situation of a cusp bifurcation, where the two saddle-node bifurcations in (a) have coalesced, for example, as another parameter of the system is varied. In this situation a small change in $\mu$ will also cause a large change in the system $F$, although it will be smooth and reversible. In Fig. 1(c), even though the saddle-node bifurcations may be outside the parameter interval of interest, abrupt large transitions in the system can result from a small

noise- or perturbation-induced change to the system even when the parameter value remains constant. It is the presence of saddle-node bifurcations in a mathematical model, even if not occurring precisely at the system's current parameter value, that are the root causes of all of the behaviors shown in Fig. 1.

    For a tipping point to be present, the underlying mathematical model will be characterized by nonlinearity, generally in the form of a positive feedback that accelerates change once change has begun. For the Arctic, one of the primary positive feedbacks

is the surface albedo. When the Arctic Ocean is frozen, the surface reflects a significant portion of the insolation back into space, but open water absorbs much more heat from the sun. Timing of the melt in the spring has significant impact (Zheng et al., 2021). An earlier melt means considerably more heat is absorbed by open water, raising the water temperature and delaying freeze up in the autumn. The freeze up date for the Beaufort, Chukchi, Laptev, and Kara Seas, for example, has been getting later by 6–11 days per decade since 1979 (Stroeve et al., 2014). September sea ice extent has been decreasing at an

accelerating rate. The linear trend from 1979 to 2001 is -7% per decade, but including data up to 2013, the linear trend is -14% per decade (Stroeve et al., 2014). Thus the observational evidence indicates that the processes behind this phenomenon are not linear at all, but nonlinear.





Past studies on GCMs have given mixed results regarding the presence of multiple stable states for ice conditions in the Arctic. Some indicate that there appears to be a continuous transition from perennial ice cover to annually ice-free that is reversible, (Schröder and Connolley, 2007; Tietsche et al., 2011; Armour et al., 2011). Other studies have shown evidence for nonlinear behavior in sea ice loss, especially in the transition from seasonally ice-free to annually ice-free, (Winton, 2006, 2008; Ridley et al., 2008). On the other hand, smaller conceptual models generally show bistability and abrupt transitions in sea ice cover, (Thorndike, 1992; Müller-Stoffels and Wackerbauer, 2011; Eisenman and Wettlaufer, 2009; Björk and Söderkvist, 2002; Abbot et al., 2011; Merryfield et al., 2008; Flato and Brown, 1996). The most common result from all these models seems to be that sea ice will likely transition from perennial to seasonally ice-free in a continuous, reversible manner, but significant warming beyond that point will likely cause an abrupt change to annually ice-free (Bathiany et al., 2016). See the introduction in Eisenman (2012). The model we present here is an annually averaged model with no seasonal component. It is not a model of sea ice in particular but rather a column model of the ocean-atmosphere coupled system that incorporates a nonlinear albedo response to surface temperature. Bistability in our model with both warm and cold solutions, corresponds to annually averaged ice-covered, or ice-free situations.

The Arctic climate model presented here is motivated by three observations. First is the observation that the climate changes taking place on the Earth today are most dramatic in the high Arctic. Therefore, it is prudent to put a special focus on understanding Arctic climate change. Second, irreversible change is inevitably the result of nonlinear geophysical processes. So, while this model is kept very simple, it does include significant nonlinear phenomena which can lead to tipping points. Third, to a fairly good approximation, the 3D spherical shell of atmosphere of the Earth is rotationally symmetric about the polar axis, if annually and zonally averaged. Due to the rotation of the Earth, Hadley, Ferrel and Polar Cells form in the global circulation. If perfect rotational symmetry is assumed, the polar axis becomes flow-invariant, and this remains approximately true for the real Earth. Thus, a 1D model restricted to the polar axis, can be expected to give useful information about climate in a neighbourhood of the pole. The study of a rotationally symmetric spherical shell model by Lewis and Langford (2008) gives support to this hypothesis. A vertical column of atmosphere at other points on Earth would have a horizontal component of velocity, invalidating the type of analysis used here. Globally averaged climate models do reduce to one (vertical) dimension, but they give little information specific to the Arctic.

The present model builds on the simple energy balance slab models of Dortmans et al. (2019), which was applied to paleoclimate transitions, and Kypke et al. (2020), which was applied to anthropogenic climate change. The primary improvement of the present model is a more physically accurate description of the atmosphere. Instead of using a slab to represent a uniform atmosphere with absorption properties similar to the real atmosphere, here we use the Schwarzschild two-stream equations to model absorption in the atmosphere explicitly as a function of altitude (Pierrehumbert, 2010, pg. 191).

A bifurcation analysis is performed on the model, tracking the steady-state solutions as carbon dioxide levels increase. The question of reversibility is a question of whether the current cold state simply warms but persists. The disappearance of this cold state through a saddle-node bifurcation would result in an abrupt change in climate that may be practically irreversible. The simpler model of Kypke et al. (2020) showed this behaviour under certain $CO_2$ representative concentration pathway scenarios. We seek here to determine if the present, more accurate model, also displays this behaviour.





Section 2 and Appendix A provide a detailed derivation of the model. The model parameter values and calibration of some of them to empirical data is presented in Appendix B. Section 3 presents the results, and the conclusions are in Section 4.

## 2 Model

The model is developed from first principles and has the following features.

- The atmosphere is a one-dimensional column at the North Pole with physical properties that vary with altitude, from the surface to the tropopause.

- The incoming solar radiation is annually averaged and undergoes reflection and absorption in the atmosphere as well as at the Earth's surface.

- The surface albedo is a nonlinear function of the surface temperature.

- A well-mixed surface boundary layer is included.

- The Earth emits longwave radiation as a black body.

- The atmosphere is considered to be a gray gas.

- The Schwarzschild two-stream equations govern the absorption and emission of both upward and downward directed longwave radiation in the atmosphere.

- The atmospheric absorption of longwave radiation is due to three factors: water vapor, $CO_2$ concentration, and clouds.

- Water vapor concentration is governed by the nonlinear Clausius-Clapeyron equation.

- Transfer of latent and sensible heat from the surface to the atmosphere is modelled.

- Both ocean and atmospheric meridional heat transport to the Arctic are dictated by empirical values.

- In the Arctic, there is a slow downward movement of air in the column corresponding to the polar circulation cell near the pole (Lewis and Langford, 2008; Langford and Lewis, 2009; Lutgens and Tarbuck, 2019). This is achieved via mass transport of air into the column in its upper portion and out of the column near the bottom.

- The radiation absorption coefficients are calibrated by fitting the model to global average data.

- The functional forms of the mass transport and atmospheric heat transport are used to calibrate the model to an empirical Arctic temperature profile.

The model domain is a vertical cylinder of cross sectional area $A$ (m$^2$) and circumference $C_b$ (m). The atmosphere is assumed to be uniform in the cross-sectional direction so that the model's dependent variables can be interpreted as cross-sectional averages that vary with the one-dimensional vertical coordinate $z \in [0, z_T]$, where $z_T$ (m) is the height of the tropopause. This domain is divided into a surface boundary layer with height $z_B$ (m), $z_B \ll z_T$, and the troposphere proper, $z \in [z_B, z_T]$. The model consists of a set of initial value problems (IVPs), with a spatial independent variable $z$ on $[0, z_B]$, that can be solved




analytically, and a two-point boundary value problem (BVP) on $[z_B, z_T]$ that depends on the solutions to the IVPs. The model has equations governing the vertical wind speed, $w$ (m s$^{-1}$), the air density, $\rho$ (kg m$^{-3}$), the upward and downward longwave radiation, $I_+$ and $I_-$ (W m$^{-2}$), the downward shortwave radiation, $I_S$ (W m$^{-2}$), the latent and sensible heat transport, $F_C$, (W

m$^{-2}$), and the temperature $T$ (K). Any reflection of shortwave radiation from either the surface or the atmosphere is ignored and is simply considered as leaving the system.

## 2.1 Mass, Momentum, and Energy Balance

The model equations in the troposphere are developed from the fundamental transport theorem in one spatial dimension:

$$\partial_t f + \partial_z \chi = S, \tag{1}$$

where $f$ is the density of some "property", $\chi$ is the flux of that property, and $S$ is a source/sink term. The time derivative term will be taken as zero since only the steady-state solution is considered. The properties subject to this equation are mass, momentum, and energy. To model the Arctic, the cylinder is centred at the north pole, and, since the atmospheric polar cell has slow downward movement near the pole, it is assumed that $w < 0$.

### 2.1.1 Mass

If the property $f$ in (1) is the mass density, $\rho$ (kg m$^{-3}$), then the flux is $\chi = \rho w$. There is mass flux across the vertical boundary of the cylinder, $M_b(z)$ ( kg m$^{-2}$ s$^{-1}$), which is assumed to be immediately spread out evenly across the layer, hence the mass flux across the vertical boundary in the model is really a mass source term in the interior giving

$$S = \frac{\text{mass entering into cylinder layer of width } \Delta z}{\text{volume of layer}} = \frac{M_b(z) C_b \Delta z}{A \Delta z} = \frac{C_b}{A} M_b(z).$$

Thus, at steady-state the mass balance equation is

130 $$\frac{d(\rho w)}{dz} = \frac{C_b}{A} M_b(z).$$

The mass flux through the vertical boundary into the column is written as

$$M_b(z) = \frac{A}{C_b(z_T - z_B)} M_{\max} \phi \left( \frac{z - z_B}{z_T - z_B} \right),$$

where $M_{\max}$ (kg m$^{-2}$ s$^{-1}$) is a nonnegative constant and $\phi(x) : [0, 1] \mapsto \mathbb{R}$ is a dimensionless function that represents the portion of inward mass flux across the vertical boundary of the column at the given altitude. Positive $\phi$ indicates inward flow.

The ratio of the cross sectional area of the column to the area of its side, $A/(C_b(z_T - z_B))$, in the definition of $M_b(z)$ is included as a useful simplifying convenience. The (positive inward) mass fluxes across the bottom and top boundaries of the column are given by $M_{\max} \Phi_B$ and $M_{\max} \Phi_T$, respectively, where $\Phi_B$ and $\Phi_T$ are dimensionless constants in the interval $[-1, 1]$. These quantities must satisfy

$$\int_0^1 \phi(x)\, dx + \Phi_B + \Phi_T = 0, \qquad \text{and} \qquad \int_0^1 \max(\phi(x), 0)\, dx + \max(\Phi_B, 0) + \max(\Phi_T, 0) = 1. \tag{2}$$


The first of these conditions dictates that there is no net mass entering the system, and the second is a normalization condition so that $M_{\max}$ represents both the total mass entering the column per unit cross sectional area, and the magnitude of the total mass leaving the column per unit cross sectional area. In order to obtain downward vertical flow throughout the cylinder, it shall be assumed that $\Phi_B < 0$, $\Phi_T > 0$, $\phi \leq 0$ in the lower part of the cylinder, and $\phi \geq 0$ in the upper part. With these definitions, the mass balance equation becomes

$$145 \quad \frac{d(\rho w)}{dz} = \frac{M_{\max}}{z_T - z_B} \phi \left( \frac{z - z_B}{z_T - z_B} \right). \tag{3}$$

### 2.1.2 Momentum

Now take the property $f$ in (1) to be the momentum density, $\rho w$. The vertical flux is $\chi = \rho w^2$, and the source term $S$ has two components, one due to contact forces (stress) (van Groesen and Molenaar, 2017, pg. 56) and one due to internal body forces (gravity):

$$150 \quad S = -\frac{dP}{dz} - \rho g,$$

where $P$ (N m$^{-2}$) is the pressure and $g$ (m s$^{-2}$) is the gravitational acceleration. It is assumed that mass entering the cylinder from the vertical boundary has no vertical momentum. Thus the momentum balance equation at steady-state is

$$\frac{d(\rho w^2)}{dz} = -\frac{dP}{dz} - \rho g. \tag{4}$$

(In the case of no flow ($w = 0$) the above would read $\frac{dP}{dz} = -\rho g$, which is the hydrostatic equation.)

### 155 2.1.3 Energy

Finally consider the case where the property in (1) is the total energy density given by

$$e = \frac{1}{2}\rho w^2 + \rho g z + c_v \rho T,$$

which corresponds to the sum of kinetic energy, gravitational potential energy, and internal heat energy densities. Here $c_v$ (J kg$^{-1}$ K$^{-1}$) is the specific heat capacity of the air. The flux has two components, one due to advection and one due to 160 conduction:

$$\chi = ew - k\frac{dT}{dz},$$

where $k$ (W m$^{-1}$ K$^{-1}$) is the thermal conductivity. The source/sink $S$ has eight terms, one due to work done by contact forces, two due to mass entering or leaving across the vertical boundary (one of these accounts for gravitational potential energy and the other internal heat energy; there is no addition to kinetic energy since the mass appearing has no velocity), three terms 165 due to radiation (shortwave downward, and longwave upward and downward), one due to latent and sensible heat transport, and one due to atmospheric heat transport. It is important to distinguish the difference between the mass transport across the boundary and the atmospheric heat transport. It is assumed that the mass moving across the vertical boundary is at the





same temperature as the mass inside at each altitude. Since mass transfer across the vertical boundary is inward in the upper portion, where the temperature is cooler, and outward in the bottom portion, mass transport results in a net transport of heat

out through the vertical boundary, but this will be small since the mass flux, $M_b(z)$, is small. The reason $M_b(z)$ is small is that it generates the average slow movement of air downward near the north pole (about 1 mm/s), due to the circulation of the polar cell. This slow averaged circulation of air does not account for the atmospheric heat transport. The main transport of heat in the atmosphere is via turbulent mixing captured in our model by a source term, $F_A(z)$ (W m$^{-3}$), whose functional form is discussed in Section A3. Thus $S$ is given by

$$S = -\frac{d(Pw)}{dz} + \frac{M_{\max}}{z_T - z_B}\phi\left(\frac{z - z_B}{z_T - z_B}\right)gz + \frac{M_{\max}}{z_T - z_B}\phi\left(\frac{z - z_B}{z_T - z_B}\right)c_v T - \frac{dI_+}{dz} + \frac{dI_-}{dz} + \frac{dI_S}{dz} - \frac{dF_C}{dz} + F_A(z).$$

The governing equations for the longwave radiation intensities are the two-stream Schwarzschild equations and for the short-wave radiation a standard absorption equation:

$$\frac{dI_+}{dz} = -\kappa\left(I_+ - \sigma T^4\right), \tag{5}$$

$$\frac{dI_-}{dz} = \kappa\left(I_- - \sigma T^4\right). \tag{6}$$

$$\frac{dI_S}{dz} = k_S \rho I_S, \tag{7}$$

where $k_S$ (m$^2$ kg$^{-1}$) is the shortwave absorption coefficient, $\sigma$ (W m$^{-2}$ K$^{-4}$) is the Stefan-Boltzmann constant, and $\kappa$ (m$^{-1}$) is the long wave absorption coefficient with terms corresponding to absorption by clouds, carbon dioxide and water vapour:

$$\kappa(\rho, T) = k_{Cl} + k_C \frac{M_{\mathrm{CO}_2}}{M_{\mathrm{A}}}\left(\frac{\mu}{10^6}\right)\rho + k_W \delta\left(\frac{z - z_B}{z_T - z_B}\right)P^{sat}(T). \tag{8}$$

Here $k_{Cl}$ (m$^{-1}$), $k_C$ (m$^2$ kg$^{-1}$), and $k_W$ (s$^2$ kg$^{-1}$) are absorption coefficients that will be calibrated, $\mu$ (ppm) is the CO$_2$

concentration expressed as the ratio of moles of CO$_2$ to moles of dry air, $M_{\mathrm{CO}_2}$ and $M_{\mathrm{A}}$ (kg mol$^{-1}$) are the molar masses of CO$_2$ and dry air, respectively, $\delta\left(\frac{z - z_B}{z_T - z_B}\right)$ is the relative humidity at altitude $z$, and $P^{sat}(T)$ (N m$^{-1}$) is the saturated water vapour partial pressure at temperature $T$. The dependence of this last quantity on $T$ is given by the Clausius-Clapeyron equation

$$P^{sat}(T) = P^{sat}(T_R)\exp\left(\frac{L_v}{R_W T_R}\frac{T - T_R}{T}\right), \tag{9}$$

where $P^{sat}(T_R)$ is the pressure at a reference temperature $T_R$ (which we take to be 273.15 K), $L_v$ (m$^2$ s$^{-2}$) is the latent heat of vaporization for water, and $R_W = R/M_W$ (J K$^{-1}$ kg$^{-1}$) is the gas constant for water, $R$ (J K$^{-1}$ mol$^{-1}$) is the universal gas constant, and $M_W$ (kg mol$^{-1}$) is the molar mass of water. The corresponding density is $\rho_W^{sat}(T) = P^{sat}(T)/(R_W T)$ by the ideal gas law. The vertical heat transport (latent and sensible heat) is assumed to be governed by a simple exponential decay

$$\frac{dF_C}{dz} = -bF_C, \tag{10}$$





where $b$ (m$^{-1}$) is a suitable decay constant. Therefore the energy balance equation at steady-state is

$$\frac{d}{dz}\left(\frac{1}{2}\rho w^3 + \rho g z w + c_v \rho T w\right) - k\frac{d^2 T}{dz^2} = -\frac{d(Pw)}{dz} + \frac{M_{\max}}{z_T - z_B}\phi\left(\frac{z - z_B}{z_T - z_B}\right)(gz + c_v T)$$
$$+ \kappa(\rho, T)\left(I_+ + I_- - 2\sigma T^4\right) + k_S \rho I_S + b F_C + F_A(z). \quad (11)$$

In order to complete the system a constitutive relation between the density $\rho$ and the pressure $P$ is needed for which we use
the ideal gas law,

$$P = R_A \rho T, \quad (12)$$

where $R_A = R/M_A$ (J kg$^{-1}$ K$^{-1}$) is the gas constant for air.

The mass, momentum, and energy balance equations, (3), (4), and (11), along with the Schwarzschild equations (5)–(6), and the equations governing shortwave absorption (7) and sensible and latent heat transport (10), are the differential equations for
the BVP on $[z_B, z_T]$. Equations (8), (9), and (12) define certain quantities in these differential equations. Before discussing the boundary conditions for the BVP it is necessary to consider the surface boundary layer.

## 2.2 Surface Boundary Layer

The model includes a boundary layer extending from $z = 0$ to $z = z_B$. It is assumed that this layer is well-mixed so that temperature $T_B = T(z_B)$ and density $\rho_B = \rho(z_B)$ in this layer are constant. The temperature of the surface, $T_S$, can in general
be larger or smaller than $T_B$.

The primary reason for including a boundary layer is a numerical one. As shown in Appendix A2, the model is numerically stiff due to the thermal conductivity of air being very small. To remove the stiffness, a limit to vanishing conduction is taken, and this results in an algebraic expression for the temperature gradient that includes the vertical wind speed as a factor in the denominator. As the vertical wind speed must be zero at the Earth's surface, there is a singularity in the temperature gradient
there. The introdcution of the surface boundary layer avoids this singularity.

The total mass crossing from the atmosphere into the boundary layer per unit time is $M_{\max}\Phi_B A$. This quantity is negative, since $\Phi_B < 0$, indicating flow out of the atmosphere and into the boundary layer. This mass exits through the vertical boundary of the layer with an assumed constant mass flux $K$, at each $z$, and conservation of mass dictates

$$\int_0^{z_B} K C_b \, dz = M_{\max}\Phi_B A \qquad \Longrightarrow \qquad K = \frac{M_{\max}\Phi_B A}{C_b z_B}.$$

($K < 0$ indicates the flux is outward.) This exiting mass carries gravitational potential energy. The change of potential energy in a slab of height $\Delta z$ at height $z$ in the boundary layer is $C_b \Delta z K g z$ so that the total change in potential energy over the boundary layer is

$$\Delta_{PE} = \int_0^{z_B} C_b K g z \, dz = \frac{1}{2}M_{\max}\Phi_B A g z_B. \quad (13)$$





Consistent with the modelling assumption that mass flux across the vertical boundary conveys no momentum or kinetic energy
to the system, the loss of mass out of the vertical boundary of the boundary layer also has no effect on the momentum or kinetic
energy. Further, since the temperature in the boundary layer, $T_B$, is assumed to be equal to the temperature of the atmosphere
at $z = z_B$, it follows that there is also no net energy change in the boundary layer due to advection of internal energy — the
internal energy entering via advection at the top of the layer is equal to the internal energy leaving the layer through the vertical
boundary.

Consider now the energy balance at the Earth's surface. There is energy transport from the surface to the boundary layer in
the form of sensible and latent heat, which is modelled, as per Pierrehumbert (2010, pgs. 396–398), as

$$F_{C0}(\rho_B, T_B, T_S) = F_C(0) = F_{\text{sensible}} + F_{\text{latent}} = c_v C_D U \rho_B (T_S - T_B) + \frac{L_v}{R_W T_B} C_D U \left( P^{sat}(T_S) - \delta(0) P^{sat}(T_B) \right), \quad (14)$$

where $C_D$ is a dimensionless drag coefficient, $U$ (m s$^{-1}$) is the horizontal wind speed, and $P^{sat}(T)$ is given by (9). Along
with this there is energy input to the surface from the sun, $I_S(0)$, some of which is reflected by the surface albedo, longwave
radiation both inward, $I_-(0)$, and outward, $I_+(0)$, and ocean heat transport, $F_O$ (W m$^{-2}$). Therefore the energy balance at the
surface is

$$F_O - I_+(0) + I_-(0) + I_S(0)(1 - \alpha(T_S)) - F_C(0) = 0, \quad (15)$$

where

$$\alpha(T_S) = \frac{1}{2} \left[ (\alpha_w + \alpha_c) + (\alpha_w - \alpha_c) \tanh \left( \frac{T_S - T_R}{T_R \omega} \right) \right] \quad (16)$$

is the surface albedo, here modelled as a sigmoid function increasing from $\alpha_c$ at cold temperatures to $\alpha_w$ at warm temperatures,
with the midway point being at the reference temperature $T_R$ (freezing point) and with a steepness of transition determined by
the dimensionless constant $\omega$.

Now consider the energy balance for the combined surface and boundary layer (one could alternatively consider just the
boundary layer without the surface, but the chosen formulation results in a slightly smaller equation). Input energy to this
combined surface and boundary layer includes ocean heat transport, and short and longwave radiation entering at $z_B$. Output
energy includes upward longwave radiation at $z_B$, the shortwave radiation reflected from the surface, and the latent and sensible
heat $F_C$ at $z_B$. Further, there are kinetic and gravitational potential energy fluxes and heat conduction in/out of the layer through
its top at $z_B$, and there is gravitational potential energy loss through the vertical boundary given by (13). Therefore the energy
density balance for the combined surface and boundary layer is

$$F_O + I_S(z_B) + I_-(z_B) - I_+(z_B) - I_S(0)\alpha(T_S) - F_C(z_B)$$
$$- \frac{1}{2} \rho_B w(z_B)^3 - \rho_B g z_B w(z_B) + k \frac{dT}{dz}(z_B) + \frac{1}{2} g M_{\max} \Phi_B z_B = 0. \quad (17)$$

Since temperature and pressure are constant in the boundary layer, the radiation equations may be solved analytically inside
the layer in order to relate the radiation terms at $z = 0$ with those at $z = z_B$. The simple ODE for $F_C$ is also easily solved in





the boundary layer. The initial (spatial independent variable $z = 0$) condition for the upward longwave radiation, $I_+(0)$, is that it is equal to the black body radiation of the surface, $\sigma T_S^4$. The initial condition for that latent heat, $F_C(0)$, is given by (14). Initial conditions for $I_-$ and $I_S$ are not necessary since only a relation between the values of these functions at 0 in terms of their value at $z_B$ is required. The IVPs for $I_+$, and $F_C$, and the ODEs for $I_-$ and $I_S$ in the boundary layer are:

$$\frac{dI_+}{dz} = -\kappa(\rho_B, T_B)(I_+ - \sigma T_B^4), \qquad\qquad I_+(0) = \sigma T_S^4,$$

$$\frac{dF_C}{dz} = -bF_C, \qquad\qquad F_C(0) = F_{C0}(\rho_B, T_B, T_S),$$

$$\frac{dI_-}{dz} = \kappa(\rho_B, T_B)(I_- - \sigma T_B^4),$$

$$\frac{dI_S}{dz} = k_S \rho_B I_S,$$

and their solutions, via standard means, give

$$I_+(z_B) = (\sigma T_S^4 - \sigma T_B^4)e^{-\kappa(\rho_B, T_B)z_B} + \sigma T_B^4, \tag{18}$$

$$F_C(z_B) = F_{C0}(\rho_B, T_B, T_S)e^{-bz_B}, \tag{19}$$

$$I_-(0) = (I_-(z_B) - \sigma T_B^4)e^{-\kappa(\rho_B, T_B)z_B} + \sigma T_B^4, \tag{20}$$

$$I_S(0) = I_S(z_B)e^{-k_S \rho_B z_B}. \tag{21}$$

Equations (18)–(19) provide two boundary conditions for the BVP on the troposphere. The energy balance equations (15), (17) along with equations (16), (20), and (21) provide two further boundary conditions.

## 2.3 Boundary Conditions for the BVP

There are eight unknown dependent variables: $w$, $\rho$, $I_+$, $I_-$, $I_S$, $F_C$, $T$, and $\frac{dT}{dz}$; the pressure $P$ can be written in terms of the others via (12). The boundary conditions for the system on the interval $[z_B, z_T]$ are wind speed at $z_B$ given by the requirement that the advected mass $Aw(z_B)\rho_B$ equals the mass flux $M_{\max}\Phi_B A$, pressure at the surface equal to the standard pressure, upward longwave radiation at $z_B$ given by (18), vertical heat transport at $z_B$ given by (19), the energy balance equations (15) and (17) with expressions from (20) and (21) substituted in, no downward longwave radiation at $z_T$, shortwave radiation at $z_T$





equal to the insolation, $Q$, less what is reflected by the clouds, $Q_R$, and a local critical point for $T$ at $z_T$:

$$\rho(z_B)w(z_B) = M_{\max}\Phi_B, \tag{22}$$

$$R_A\rho(z_B)T(z_B) = P_0, \tag{23}$$

$$I_+(z_B) = \left(\sigma T_S^4 - \sigma T(z_B)^4\right)e^{-\kappa(\rho(z_B),T(z_B))z_B} + \sigma T(z_B)^4, \tag{24}$$

$$F_C(z_B) = F_{C0}(\rho(z_B),T(z_B),T_S)e^{-bz_B}, \tag{25}$$

$$0 = F_O - \sigma T_S^4 + \left(I_-(z_B) - \sigma T(z_B)^4\right)e^{-\kappa(\rho(z_B),T(z_B))z_B} + \sigma T(z_B)^4$$
$$+ I_S(z_B)e^{-k_S\rho(z_B)z_B}(1-\alpha(T_S)) - F_{C0}(\rho(z_B),T(z_B),T_S), \tag{26}$$

$$0 = F_O - I_+(z_B) + I_+(z_B) + I_S(z_B) - I_S(z_B)e^{-k_S\rho(z_B)z_B}\alpha(T_S) - F_C(z_B)$$
$$+ k\frac{dT}{dz}(z_B) - \frac{1}{2}\rho(z_B)w(z_B)^3 - \rho(z_B)gz_Bw(z_B) + \frac{1}{2}gM_{\max}\Phi_Bz_B, \tag{27}$$

$$I_-(z_T) = 0, \tag{28}$$

$$I_S(z_T) = Q - Q_R, \tag{29}$$

$$\frac{dT}{dz}(z_T) = 0, \tag{30}$$

where $F_{C0}$ is given by (14). The last three terms of Eqn. (27) may be simplified using Eqn. (22) so that they read

$$-\frac{M_{\max}\Phi_Bw(z_B)^2}{2} - \frac{M_{\max}\Phi_Bgz_B}{2}$$

There are nine boundary conditions, but there are only eight dependent variables for which we have differential equations. The discrepancy is explained by the presence of $T_S$, which is an additional scalar variable. The nine boundary conditions determine eight conditions for the differential equations as well as the value for $T_S$. One way of treating this is simply to extend the system of differential equations to include the equation

$$\frac{dT_S}{dz} = 0. \tag{31}$$

In addition, as described in Appendix A, to avoid numerical stiffness we take the limit as the heat conduction of air, $k$, tends to zero. This effectively reduces the size of the model by one dimension.

The model is nondimensionalized and put in standard form as detailed in Appendix A. The parameter values and their calibration to empirical data is provided in Appendix B.

## 3 Results

For the Arctic parameter values given in Appendix B and for a given $CO_2$ concentration, $\mu$, the model can be solved numerically. We used MATLAB's builtin BVP solver "`bvp5c`" to solve individual instances of the (nondimensionalized) model, and AUTO for continuation calculations. The results of the model for $\mu = 390$ ppm are shown in Figure 2. The altitude dependence of the mass flux $\phi$ and the atmospheric meridional heat transport, $F_A$, were calibrated to an empirical Arctic temperature





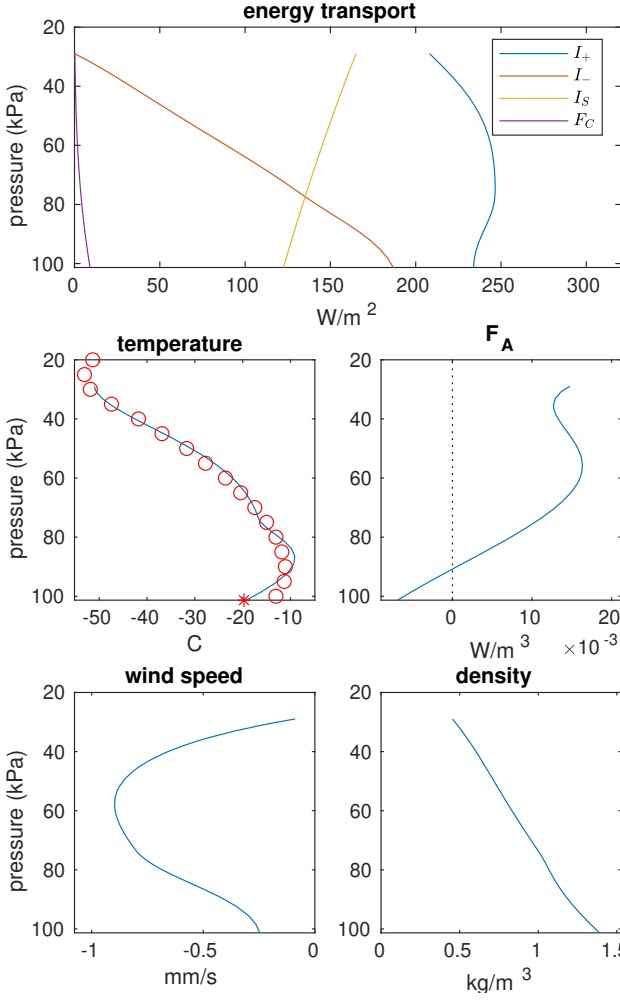

**Figure 2.** Results of the fully calibrated Arctic model at $\mu = 390$ ppm. The red circles in the middle left plot are the data from Cronin and Jansen (2016). The red asterisk in the same plot is the surface temperature, $T_S$.


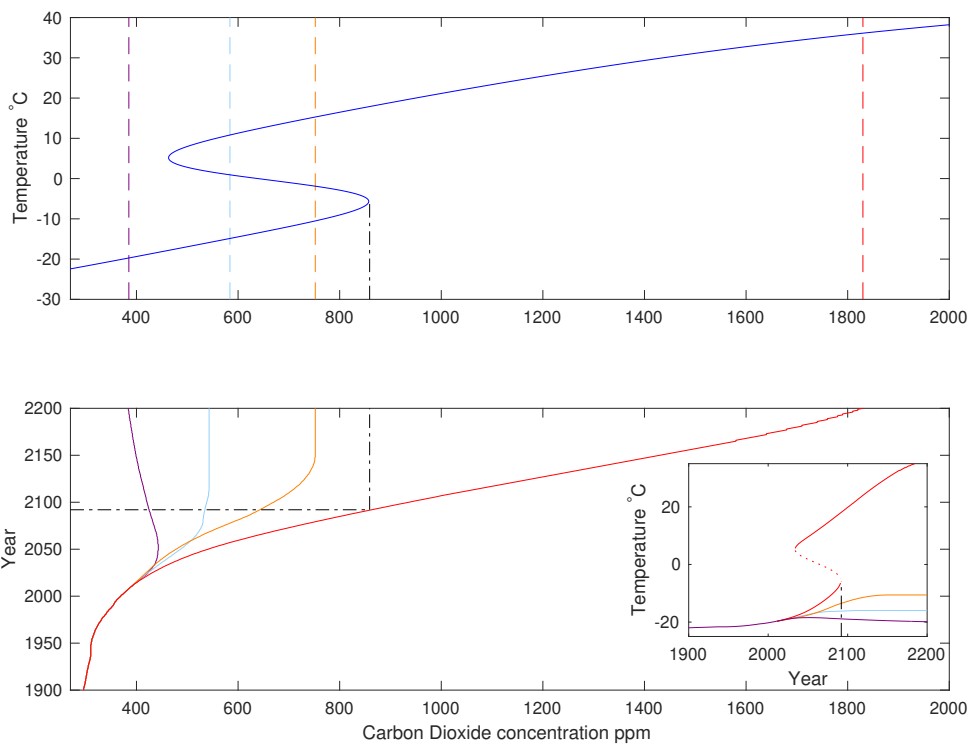

**Figure 3.** Top panel: Model surface temperature as a function of $CO_2$ concentration, $\mu$. Bottom panel: $CO_2$ concentration levels for RCPs 2.6, 4.5, 6.0, and 8.5 (left to right). The dashed lines in the upper panel correspond to the $CO_2$ concentration levels for the four RCPs in the year 2200. The dotted line extends from the bifurcation point in the upper panel to RCP 8.5 in the lower panel indicating the bifurcation occurs in approximately the year 2100 for this scenario. The inset shows the predicted surface temperature as a function of the year for the four RCPs.

profile from Cronin and Jansen (2016) as detailed in Appendix B. From this figure we see that the model fits the temperature

profile very well, with some discrepancy near the surface. The overall atmospheric heat transport (middle right plot of Fig. 2) indicates that, for our model the upper half of the troposphere receives the most input of heat, while the bottom fifth actually has a net outward heat transport. This removal of heat near the bottom is likely the cause of the discrepancy between our model values and the Cronin and Jansen data. The negative values of $F_A$ near the surface are due to our modelling choice for $F_A$. It is possible that alternative modelling formulations for $F_A$ could yield a better fit to the Cronin and Jansen data, however,

the forms that we tried (including the ones reported in Appendix B and a few others) did not improve upon the one used here. Nonetheless, given the relative simplicity of our model, we feel the fact that it can fit the data as well as it does is remarkable.

For $\mu = 390$, the surface temperature from the solved model is -19.7 C. Starting from this solution we numerically continued the solution to other values of $\mu$, resulting in the S-shaped curve in the top panel of Fig. 3. Between approximately $\mu = 464$ and $\mu = 859$ there are three solutions. The lower and upper solutions are stable cold and warm solutions, while the middle




branch is unstable. Currently the Arctic is on the cold branch of this curve. The model predicts that as $CO_2$ levels rise, the
equilibrium surface temperature in the Arctic will increase gradually at first. However, when $\mu$ exceeds 859 ppm, the model
displays a saddle-node bifurcation where the stable cold solution is annihilated together with the unstable solution. At this
point the climate would rapidly approach the warm stable solution where surface temperatures are significantly higher. The
temperatures on the warm branch may seem unreasonably high. Although our model includes specific details of some of the
natural phenomena governing the Arctic climate behavior, and is calibrated with real data, it is primarily a qualitative model
rather than quantitative. It is the fact that the model predicts the qualitative feature of a saddle-node bifurcation that is the
important result, not the precise temperature of the model's warm solution. Part of the reason the model has relatively high
temperatures on the warm branch is because it has constant values for $F_O$ and $F_A^{\text{tot}}$. It is likely that should the Arctic's annually
averaged temperature rise dramatically, both $F_O$ and $F_A^{\text{tot}}$ would be affected in a downward direction, which would reduce the
warm equilibrium temperatures to some degree. Another important emergent feature of the model is that there is bistability
for $\mu$ in the range $[464, 859]$. Thus, even though the Arctic may in the future be on the cold branch in this range, it is possible
that a strong enough climate disturbance could push the climate out of the basin of attraction of the cold solution and into
the basin of attraction of the warm solution. The necessary strength of such a disturbance decreases as one moves closer to
the upper end of this range. Our model does not incorporate the seasonal variation of solar input to the Arctic, but rather uses
an annually averaged value. Thus the equilibria in our model are annual averages. The seasonal variation of insolation would
effectively result in oscillations around the annual average. These oscillations themselves may make a significant contribution
to the "disturbance" needed to push the system out of the basin of attraction of the cold branch.

   The International Panel on Climate Change (IPCC) has published various $CO_2$ emission scenarios for the future based on
possible levels of global action to suppress such emissions (Intergovernmental Panel on Climate Change, 2013, Box TS.6), (van
Vuuren et al., 2011). These scenarios are called Representative Concentration Pathways (RCPs) and are numbered based on
the radiative forcing in the year 2100 due to anthropogenic emissions compared to the year 1750. The original four published
RCPs are RCP 2.6, RCP 4.5, RCP 6.0, and RCP 8.5, representing strong mitigation (2.6) through "ignore the problem" (8.5)
responses by world governments. Each RCP indicates likely $CO_2$ concentration levels in the atmosphere out to the year 2100.
From 2100 to 2200, the scenarios assume a "constant composition commitment," which essentially freezes emission levels and
eventually leads to a constant $CO_2$ level in the atmosphere for all but RCP 8.5. These RCPs are plotted in the lower panel of
Fig. 3. The $CO_2$ levels for the four different pathways at the year 2200 are continued as dashed lines into the upper panel.
It is clear from the figure that RCP 8.5 leads to $CO_2$ concentrations that far surpass the saddle-node bifurcation, whereas the
other three RCPs do not. This result is in agreement with our simpler model (Kypke et al., 2020). Both RCP 6.0 and RCP 4.5
end at levels within the bistable range and indeed all the RCPs except RCP 2.6 are within that range from about the year 2050
onward. The black dotted line extending from the bifurcation in the upper panel to RCP 8.5 in the lower panel illustrates that
RCP 8.5 reaches the bifurcation near the year 2092.

   The curve of equilibria in the upper panel of Fig. 3 displays hysteresis: $CO_2$ levels rising past 859 ppm will cause a jump
from the cold equilibrium state to the warm state, but a return to the cold state will not happen until $CO_2$ levels are brought
below 464 ppm, where the saddle-node bifurcation of the warm equilibrium is located (left bend of the S-curve). If $CO_2$ levels

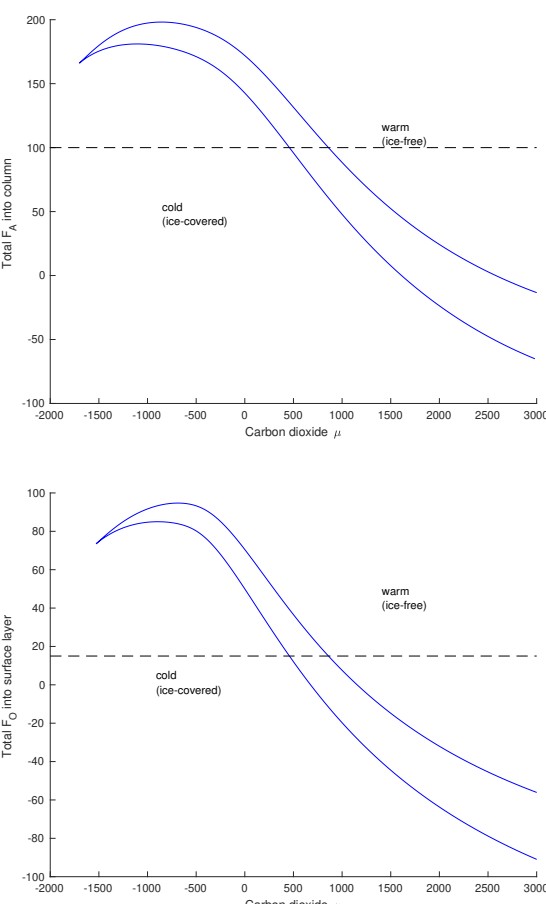

**Figure 4.** Top panel: Bifurcation diagram showing location of saddle-node bifurcations as $F_A^{\text{tot}}$ and $\mu$ are varied. The dashed line indicates the value of $F_A^{\text{tot}}$ used in the model. The narrow area between the curves is the region of bistability. Abrupt transition from a cold state to a warm state occurs on the right curve, while abrupt transition from a warm to a cold state occurs at the left curve. Bottom panel: Same except $F_O$ is the varying parameter on the vertical axis rather than $F_A^{\text{tot}}$.

follow a trajectory similar to RCP 8.5, Arctic climate may change drastically in less than 100 years, but a return to the current cold state may be essentially impossible for thousands of years afterward, assuming humankind can develop and implement the required technology to reduce atmospheric carbon levels sufficiently.

Total atmospheric heat transport, $F_A^{\text{tot}}$, and ocean heat transport, $F_O$, are inputs to the model that are empirically based and are not altered by the state of the model itself. We investigated the persistence of the saddle-node bifurcation and hysteresis

in the presence of the uncertainty of these two values. Figure 4 shows two two-dimensional bifurcation diagrams plotting the locations of the saddle-node bifurcations in the model (the left and right bends of the S-curve in Fig. 3). The top panel of Fig. 4 shows the bifurcations as both $F_A^{\text{tot}}$ and $\mu$ vary. The bottom panel shows a similar diagram where $F_O$ replaces $F_A^{\text{tot}}$.




The right curve in each of the panels is a curve of saddle-node bifurcations corresponding to the right bend in the S-curve in Fig. 3; they are transitions from a cold state to a warm state. Similarly, the left curve in each of the panels is a curve of

saddle-node bifurcations that correspond to the left bend in the S-curve, and are the transitions from a warm state to a cold state. The narrow area between the two curves is the region of bistability. The curves meet in a cusp bifurcation, but it is important to note that this only occurs when $CO_2$ levels are mathematically negative. Thus the two saddle-node bifurcations and the hysteresis phenomenon will be present for all physically possible $CO_2$ levels, and for all reasonable values of $F_A^{\mathrm{tot}}$ and $F_O$. These bifurcation curves also make it evident that a transition from a cold to a warm state occurs if either $F_A^{\mathrm{tot}}$ or $F_O$ are

increased, even if $CO_2$ levels are constant. If $F_A^{\mathrm{tot}}$ and $F_O$ increase along with carbon dioxide levels in the near future, which seems reasonably likely, then the saddle-node bifurcation will occur at lower levels of $CO_2$. For example, a ten percent increase of $F_A^{\mathrm{tot}}$ to 110 W m$^{-2}$ from 100 would cause the saddle-node bifurcation to occur at about $\mu = 754$ ppm, which would mean that both RCP8.5 and RCP6.0 would pass through the saddle-node bifurcation.

## 4 Conclusions

Although the model presented here is clearly a simplification of the climate, made possible by the near invariance of the vertical flow on the polar axis, we believe it captures some of the most important aspects relevant for Arctic climate change. The model predicts that if humanity keeps carbon emission levels close to RCP 6.0 or lower, then the Arctic will not likely undergo a sudden dramatic rise in annual average temperature. However, if carbon emissions are much worse than RCP 6.0, such a change is likely, and the cause is a saddle-node bifurcation of the stable cold equilibrium. Such a change would clearly have

catastrophic effects on the Arctic environment leading to massive global effects. These results are in agreement with those of Årthun et al. (2021) who, from a study of various CMIP6 models of Arctic climate, predict that under a low emissions scenario sea ice loss will be seasonal, but for a high emissions scenario it will be year-round for all areas of the Arctic. Further, the hysteresis displayed by the model indicates that a change of this nature may be practically irreversible. Although some comfort may be taken that only the worst of the four carbon pathway scenarios ends in such a catastrophic change, the model shows

that both RCP 6.0 and RCP 4.5 lie in the region of bistability from the year 2070 onward. In a bistable situation, external disturbances could cause the system to jump into the basin of attraction of the warm equilibrium, effectively bringing about the catastrophic change prior to the bifurcation. The model is too simple to allow for any reasonable measure of the likelihood of such an occurrence, but the important thing is that the model exhibits bistability in the parameter region where the system is likely to reside within 50 year's time. This bistability has been shown to persist regardless of the values of the two biggest

"unknowns" in the model, the atmospheric and ocean heat transport to the Arctic from the mid-latitudes.

Our model addresses the equilibrium state only and represents the Arctic temperature as an annual average. The real Arctic climate undergoes massive seasonal changes, which effectively means that the system is actually oscillating around the equilibrium temperatures of our model. Fig. 3 suggests that such temperature oscillations would not likely need to be very large to push a system located on a cold solution in the bistable regime to "above" the unstable solution and so into the basin of

attraction of the warm solution. Seasonal variations in Arctic temperatures and sea ice are studied in many works including





Eisenman and Wettlaufer (2009) who argue that a tipping point to a year-round ice-free Arctic is not likely to occur while the Arctic is ice-covered for a significant portion of the year, but once the Arctic is seasonally ice-free for a sufficient number of months in a year, such a tipping point becomes more likely. A possible enhancement to our model would be to include seasonal solar variation and an ice model like in Eisenman and Wettlaufer (2009).

**Appendix A: Model Derivation Details**

This appendix provides details regarding the model. The model is written in a nondimensional form in Section A1, defining nondimensional variables and parameters. This system is then transformed in Section A2 to a standard form of nine first order ordinary differential equations and corresponding boundary conditions. This standard form makes it evident that the system is numerically stiff due to the fact that the thermal conductivity, $k$, of air is very small. To remove this stiffness, the limit as $k \to 0$ is applied, which reduces the system by one dimension. Section A3 discusses our choice of the functional forms for the dependence of relative humidity, atmospheric meridional heat transport, and mass flux on altitude.

**A1    Non-Dimensionalization**

Define the non-dimensional variables

$$
\begin{aligned}
&\hat{z} = \frac{z - z_B}{z_T - z_B}, &&y_1 = \frac{w c_v \rho_0 T_R}{\sigma T_R^4}, &&y_2 = \frac{\rho}{\rho_0}, &&y_3 = \frac{I_+}{\sigma T_R^4}, &&y_4 = \frac{I_-}{\sigma T_R^4}, \\
&y_5 = \frac{I_S}{\sigma T_R^4}, &&y_6 = \frac{F_C}{\sigma T_R^4}, &&y_7 = \frac{T}{T_R}, &&y_8 = \left( \frac{z_T - z_B}{T_R} \right) \frac{dT}{dz}, &&y_9 = \frac{T_S}{T_R},
\end{aligned} \tag{A1}
$$

where $P_0$ (N m$^{-2}$) is the standard atmospheric pressure, $T_R$ is the freezing point of water in Kelvin, $\rho_0 = P_0/(R_A T_R)$ is the density at standard pressure and freezing temperature, and $\sigma T_R^4/(c_v \rho_0 T_R)$ represents the vertical velocity required to move a parcel of air with standard density at freezing temperature so that the power transferred is equal to the radiative power for a black body at the same temperature. This comes to be about 1 mm/s.





Applying the change of variables to the troposphere BVP, (3)–(7), (10), (11), (31), we get

$$\frac{d}{d\hat{z}}(y_1 y_2) = D\phi(\hat{z}), \tag{A2}$$

$$H\frac{d}{d\hat{z}}(y_1^2 y_2) = -J\frac{d}{d\hat{z}}(y_2 y_7) - E y_2, \tag{A3}$$

$$\frac{dy_3}{d\hat{z}} = -\hat{\kappa}(y_2, y_7)\left(y_3 - y_7^4\right), \tag{A4}$$

$$\frac{dy_4}{d\hat{z}} = \hat{\kappa}(y_2, y_7)\left(y_4 - y_7^4\right), \tag{A5}$$

$$\frac{dy_5}{d\hat{z}} = G_S y_2 y_5, \tag{A6}$$

$$\frac{dy_6}{d\hat{z}} = -B_1 y_6, \tag{A7}$$

$$\frac{dy_7}{d\hat{z}} = y_8, \tag{A8}$$

$$\epsilon\frac{dy_8}{d\hat{z}} = \frac{d}{d\hat{z}}\left(\frac{H}{2}y_1^3 y_2 + E y_1 y_2 \hat{z} + y_1 y_2 y_7\right) + J\frac{d}{d\hat{z}}(y_1 y_2 y_7) - DE\hat{z}\phi(\hat{z}) - D\phi(\hat{z})y_7$$

$$- \hat{\kappa}(y_2, y_7)\left(y_3 + y_4 - 2y_7^4\right) - G_S y_2 y_5 - B_1 y_6 - \hat{F}_A(\hat{z}), \tag{A9}$$

$$\frac{dy_9}{d\hat{z}} = 0, \tag{A10}$$

for $\hat{z} \in [0, 1]$. The boundary conditions, (22)–(30), become

$$y_1(0)y_2(0) = D\Phi_B, \tag{A11}$$

$$y_2(0)y_7(0) = 1, \tag{A12}$$

$$y_3(0) = \left(y_9(0)^4 - y_7(0)^4\right)e^{-\hat{\kappa}(y_2(0),y_7(0))\zeta} + y_7(0)^4, \tag{A13}$$

$$y_6(0) = \hat{F}_{C0}(y_2(0), y_7(0), y_9(0))e^{-B_1\zeta}, \tag{A14}$$

$$0 = F - y_9(0)^4 + \left(y_4(0) - y_7(0)^4\right)e^{-\hat{\kappa}(y_2(0),y_7(0))\zeta} + y_7(0)^4$$

$$+ y_5(0)e^{-G_S y_2(0)\zeta}(1 - \hat{\alpha}(y_9(0))) - \hat{F}_{C0}(y_2(0), y_7(0), y_9(0)), \tag{A15}$$

$$0 = F - y_3(0) + y_4(0) + y_5(0) - y_5(0)e^{-G_S y_2(0)\zeta}\hat{\alpha}(y_9(0)) - y_6(0) + \epsilon y_8(0)$$

$$- \frac{1}{2}HD\Phi_B y_1(0)^2 - \frac{1}{2}DE\Phi_B\zeta, \tag{A16}$$

$$y_4(1) = K_-, \tag{A17}$$

$$y_5(1) = K_S, \tag{A18}$$

$$y_8(1) = 0, \tag{A19}$$





where

$$\hat{F}_{C0}(y_2, y_7, y_9) = B_2 y_2 (y_9 - y_7) + \frac{B_3}{y_7}\left[\exp\left(G_{W1}\left(1 - \frac{1}{y_9}\right)\right) - \delta(0)\exp\left(G_{W1}\left(1 - \frac{1}{y_7}\right)\right)\right], \tag{A20}$$

$$\hat{\kappa}(y_2, y_7) = \kappa(y_2 \rho_0, y_7 T_R)(z_T - z_B) = G_{Cl} + G_C \hat{\mu} y_2 + \frac{G_{W2}\delta(\hat{z})}{y_7}\exp\left(G_{W1}\left(1 - \frac{1}{y_7}\right)\right), \tag{A21}$$

$$\hat{F}_A(\hat{z}) = \frac{(z_T - z_B)F_A(\hat{z}(z_T - z_B) + z_B)}{\sigma T_R^4}, \tag{A22}$$

$$\hat{\alpha}(y_9) = \frac{1}{2}\left[(\alpha_w + \alpha_c) + (\alpha_w - \alpha_c)\tanh\left(\frac{y_9 - 1}{\omega}\right)\right], \tag{A23}$$

are nondimensionalized functions describing the sensible/latent heat flux from the surface, the absorption of long wave radiation due to greenhouse gases, the atmospheric heat transport, and the surface albedo; and $\alpha_w$, $\alpha_c$, $\omega$, $\Phi_B$, and

$$\begin{aligned}
& B_1 = b(z_T - z_B), && B_2 = \frac{c_v \rho_0 C_D U}{\sigma T_R^3}, && B_3 = \frac{\rho_W^{sat}(T_R)C_D U L_v}{\sigma T_R^4}, \\
& D = \frac{M_{\max}c_v}{\sigma T_R^3}, && E = \frac{g(z_T - z_B)}{c_v T_R}, && F = \frac{F_O}{\sigma T_R^4}, \\
& G_{Cl} = k_{Cl}(z_T - z_B), && G_C = k_C \frac{M_{CO_2}}{M_A}\rho_0(z_T - z_B), && G_{W1} = \frac{L_v}{R_W T_R}, \\
& G_{W2} = k_W \rho_W^{sat}(T_R)(z_T - z_B), && G_S = k_S \rho_0(z_T - z_B), && H = \frac{\sigma^2 T_R^5}{c_v^3 \rho_0^2}, \\
& J = \frac{R_A}{c_v}, && K_- = \frac{I_-^{TP}}{\sigma T_R^4}, && K_S = \frac{I_S^{TP} - Q_R}{\sigma T_R^4}, \\
& \epsilon = \frac{k}{(z_T - z_B)\sigma T_R^3}, && \hat{\mu} = \frac{\mu}{10^6}, && \zeta = \frac{z_B}{z_T - z_B},
\end{aligned} \tag{A24}$$

are nondimensional constants.

## A2  Standard Form and Vanishing Conduction Limit

The BVP given by (A2)–(A19) can be put in standard form via algebraic manipulations. First we expand the derivative on the left side of (A3) and use (A2) to simplify it:

$$\frac{d}{d\hat{z}}(y_1^2 y_2) = y_1 \frac{d}{d\hat{z}}(y_1 y_2) + (y_1 y_2)\frac{dy_1}{d\hat{z}} = y_1 D\phi(\hat{z}) + y_1 y_2 \frac{dy_1}{d\hat{z}}.$$

Thus (A3) may be replaced with

$$H y_1 y_2 \frac{dy_1}{d\hat{z}} + J y_7 \frac{dy_2}{d\hat{z}} = -J y_2 y_8 - H D y_1 \phi(\hat{z}) - E y_2, \tag{A25}$$

where we have used (A8) to replace the derivative of $y_7$. We can also expand the derivative in (A2) to get the equivalent equation

$$y_2 \frac{dy_1}{d\hat{z}} + y_1 \frac{dy_2}{d\hat{z}} = D\phi(\hat{z}). \tag{A26}$$





Equations (A25) and (A26) are a linear system in $\frac{dy_1}{d\hat{z}}$ and $\frac{dy_2}{d\hat{z}}$, namely

$$
\begin{bmatrix} Hy_1y_2 & Jy_7 \\ y_2 & y_1 \end{bmatrix} \begin{bmatrix} \frac{dy_1}{d\hat{z}} \\ \frac{dy_2}{d\hat{z}} \end{bmatrix} = \begin{bmatrix} -Jy_2y_8 - HDy_1\phi(\hat{z}) - Ey_2 \\ D\phi(\hat{z}) \end{bmatrix}.
$$

Solving this yields

$$
\frac{dy_1}{d\hat{z}} = \frac{(Jy_7 + Hy_1^2)D\phi(\hat{z}) + y_1y_2(E + Jy_8)}{y_2(Jy_7 - Hy_1^2)}, \tag{A27}
$$
$$
\frac{dy_2}{d\hat{z}} = -\frac{2Hy_1D\phi(\hat{z}) + y_2(E + Jy_8)}{Jy_7 - Hy_1^2}. \tag{A28}
$$

Now expanding the derivatives on the right hand side of (A7) and simplifying we obtain

$$
\epsilon\frac{dy_8}{d\hat{z}} = \left[H\frac{y_1^2}{2} + \frac{J^2y_7^2 + H^2y_1^4}{Jy_7 - Hy_1^2}\right]D\phi(\hat{z}) + (J+1)y_1y_2y_8 + Jy_1y_2\frac{Ey_7 + Hy_1^2y_8}{Jy_7 - Hy_1^2}
$$
$$
- \hat{\kappa}(y_2, y_7)y_2\left(y_3 + y_4 - 2y_7^4\right) - G_Sy_2y_5 - B_1y_6 - \hat{F}_A(\hat{z}). \tag{A29}
$$

In summary the BVP for the troposphere in standard form is given by

$$
\frac{dy_1}{d\hat{z}} = \frac{(Jy_7 + Hy_1^2)D\phi(\hat{z}) + y_1y_2(E + Jy_8)}{y_2(Jy_7 - Hy_1^2)}, \tag{A30}
$$
$$
\frac{dy_2}{d\hat{z}} = -\frac{2Hy_1D\phi(\hat{z}) + y_2(E + Jy_8)}{Jy_7 - Hy_1^2} \tag{A31}
$$
$$
\frac{dy_3}{d\hat{z}} = -\hat{\kappa}(y_2, y_7)\left(y_3 - y_7^4\right), \tag{A32}
$$
$$
\frac{dy_4}{d\hat{z}} = \hat{\kappa}(y_2, y_7)\left(y_4 - y_7^4\right), \tag{A33}
$$
$$
\frac{dy_5}{d\hat{z}} = G_Sy_2y_5, \tag{A34}
$$
$$
\frac{dy_6}{d\hat{z}} = -B_1y_6, \tag{A35}
$$
$$
\frac{dy_7}{d\hat{z}} = y_8, \tag{A36}
$$
$$
\epsilon\frac{dy_8}{d\hat{z}} = \left[H\frac{y_1^2}{2} + \frac{J^2y_7^2 + H^2y_1^4}{Jy_7 - Hy_1^2}\right]D\phi(\hat{z}) + (J+1)y_1y_2y_8 + Jy_1y_2\frac{Ey_7 + Hy_1^2y_8}{Jy_7 - Hy_1^2}
$$
$$
- \hat{\kappa}(y_2, y_7)\left(y_3 + y_4 - 2y_7^4\right) - G_Sy_2y_5 - By_6 - \hat{F}_A(\hat{z}), \tag{A37}
$$
$$
\frac{dy_9}{d\hat{z}} = 0, \tag{A38}
$$





with boundary conditions

$$y_1(0)y_2(0) = D\Phi_B, \tag{A39}$$

$$y_2(0)y_7(0) = 1, \tag{A40}$$

$$y_3(0) = \left(y_9(0)^4 - y_7(0)^4\right)e^{-\hat{\kappa}(y_2(0),y_7(0))\zeta} + y_7(0)^4, \tag{A41}$$

$$y_6(0) = \hat{F}_{C0}(y_2(0),y_7(0),y_9(0))e^{-B_1\zeta}, \tag{A42}$$

$$0 = F - y_9(0)^4 + \left(y_4(0) - y_7(0)^4\right)e^{-\hat{\kappa}(y_2(0),y_7(0))\zeta} + y_7(0)^4$$
$$+ y_5(0)e^{-G_S y_2(0)\zeta}(1 - \hat{\alpha}(y_9(0))) - \hat{F}_{C0}(y_2(0),y_7(0),y_9(0)), \tag{A43}$$

$$0 = F - y_3(0) + y_4(0) + y_5(0) - y_5(0)e^{-G_S y_2(0)\zeta}\hat{\alpha}(y_9(0)) - y_6(0) + \epsilon y_8(0)$$
$$- \frac{1}{2}HD\Phi_B y_1(0)^2 - \frac{1}{2}DE\Phi_B\zeta, \tag{A44}$$

$$y_4(1) = K_-, \tag{A45}$$

$$y_5(1) = K_S, \tag{A46}$$

$$y_8(1) = 0. \tag{A47}$$

Of the nondimensional constants all are order one except $H = 7.96 \times 10^{-12}$ and $\epsilon = 2.3 \times 10^{-6}$ (with $z_T = 9000$). But the constant $H$ occurs in the system only in summations with other non-derivative terms, hence the fact it is small only means that those terms contribute little. It does not cause stiffness. The constant $\epsilon$ however, does cause stiffness in Eqn. (A37) of the system, and $y_8$, which is the dimensionless rate of vertical temperature change $dT/dz$, will approach the $y_8$-nullcline very rapidly (which means in a very short $z$-distance from either boundary). To simplify numerical computation we take the limit as $\epsilon$ goes to zero, which is equivalent to saying that conduction is negligible. In this limit, (A37) is an algebraic expression from which we can isolate $y_8$:

$$y_8(\hat{z}) = -\left[\left(H\frac{y_1^2}{2} + \frac{J^2 y_7^2 + H^2 y_1^4}{Jy_7 - Hy_1^2}\right)D\phi(\hat{z}) + \frac{EJy_1y_2y_7}{Jy_7 - Hy_1^2} - G_S y_2 y_5\right.$$
$$\left. -\hat{\kappa}(y_2,y_7)\left(y_3 + y_4 - 2y_7^4\right) - By_6 - \hat{F}_A(\hat{z})\right]\left[(J+1)y_1y_2 + \frac{JHy_1^3 y_2}{Jy_7 - Hy_1^2}\right]^{-1}. \tag{A48}$$

With this expression for $y_8$ the system is reduced by eliminating Eqn. (A37) and boundary condition (A47). Since $y_1$, the dimensionless vertical velocity, is a factor in the denominator it is necessary that $y_1$ be nonzero throughout the atmosphere in order not to introduce a singularity. For this reason we select $\Phi_T > 0$, $\Phi_B < 0$ and $\phi$ such that it is nonnegative in the upper atmosphere and nonpositive in the lower atmosphere.





### A3 Modelling Choices for Functional Forms

#### A3.1 Relative humidity

The relative humidity is modelled as a linear function decreasing with altitude from a higher surface value, $\delta_B$, to a lower value at the tropopause, $\delta_T$. Specifically,

$$\delta(\hat{z}) = \delta_B(1 - \hat{z}) + \delta_T \hat{z}. \tag{A49}$$

#### A3.2 Atmospheric heat transport

Atmospheric heat transport is primarily due to large scale turbulent mixing of the column with its environment. This mixing is not modelled explicitly, but the thermal energy supplied to the column by it is represented by the function $F_A(z)$. The integral of $F_A(z)$ over the atmosphere thickness represents the total atmospheric heat transport in/out of the system. So given a set amount of such energy, $F_A^{\text{tot}}$ we have

$$F_A^{\text{tot}} = \int_0^{z_T} F_A(z)\, dz = \sigma T_R^4 \int_0^1 \hat{F}_A(\hat{z})\, d\hat{z}.$$

This provides one restriction on the function $F_A(z)$, but its precise form is not prescribed.

Getting reasonable results from the model seems to require a delicate balance for $F_A$, especially near the tropopause. If it is too small, then the temperature drops precipitously toward absolute zero; if it is too big, the temperature turns around and climbs rapidly. This behaviour is sensitive to $F_A(z)$ for $z$ near $z_T$, that is, $\hat{z}$ near 1. In order to automate the choice of $F_A(z)$ we have proceeded as follows:

1. Choose $\hat{F}_A(1)$ such that the temperature gradient at the tropopause is zero, that is, re-impose boundary condition (A47). Thus use (A48) with $y_8(1) = 0$, to solve for $\hat{F}_A(1)$. Let $\hat{F}_{A0}$ denote this value of $\hat{F}_A(1)$, and let $F_{A0}$ denote the corresponding dimensionful value of $F_A(z_T)$, that is, $F_{A0} = \hat{F}_{A0}\sigma T_R^4/(z_T - z_B)$.

2. Assume that $F_A(z)$ is of the form

$$F_A(z) = F_{Ab}(z) + \frac{F_{A\max}}{z_T - z_B}\psi\left(\frac{z - z_B}{z_T - z_B}\right),$$

where the base function $F_{Ab}(z)$ is a linear function passing through zero at the midpoint of the atmosphere and equal to $F_{A0}$ at $z_T$, that is,

$$F_{Ab}(z) = F_{A0}\frac{2z - (z_T + z_B)}{z_T - z_B}, \qquad \Longleftrightarrow \qquad \hat{F}_{Ab}(\hat{z}) = \hat{F}_{A0}(2\hat{z} - 1).$$

This base portion of $F_A$ contributes no net heat to the column, it is simply a factor that essentially moves heat around in the column in order to assure the temperature gradient at the top is zero.





The remaining portion of $F_A$ is the actual atmospheric heat transport entering the column from outside. We assume that $\psi(\hat{z})$ satisfies

$$\psi(1) = 0 \qquad \text{and} \qquad \int_0^1 \psi(x)\,dx = 1,$$

so that the value of $F_A$ is not altered at the tropopause, and so that $F_{A\max}$ (W m$^{-2}$) represents the total energy flux of atmospheric heat transport entering the column. Thus the non-dimensional function is

$$\hat{F}_A(\hat{z}) = \hat{F}_{A0}(2\hat{z} - 1) + \hat{F}_{A\max}\psi(\hat{z}),$$

where $\hat{F}_{A\max} = \frac{F_{A\max}}{\sigma T_R^4}$.

3. As a numerical issue, since the boundary condition value $\hat{F}_{A0}$ is needed in the computation of the vector field, and since the MATLAB solver we are using does not have a way of making boundary condition information available to the vector field computation function, we circumvented this issue by adding another variable to the problem $y_{10}$, with differential equation $\frac{dy_{10}}{d\hat{z}} = 0$ and boundary condition $y_{10}(1) = \hat{F}_{A0}$.

4. Choice of the functional form of $\psi(\hat{z})$ is somewhat open, we tested the following two forms:

$$\psi(\hat{z}) = g_1(1 - \hat{z}, L_\psi), \tag{A50}$$
$$\psi(\hat{z}) = g_2(1 - \hat{z}, L_\psi), \tag{A51}$$

where the functions $g_1$ and $g_2$ are defined as

$$g_1(x, L) = \frac{L\pi}{1 - \cos(L\pi)}\sin(L\pi x),$$
$$g_2(x, L) = \frac{2L\pi}{2L\pi - \sin(2L\pi)}\left(1 - \cos(2L\pi x)\right), \tag{A52}$$

and where $L$ is a parameter in $(0, 1]$ free to be chosen. (The functions $g_1$ and $g_2$ will also be utilized in the modelling of $\phi(\hat{z})$.) The primary difference between these two forms is that the first has a non-zero slope at $x = 0$, while the latter has a zero slope there.

### A3.3 Mass flux

The function $\phi$ dictates the mass flux across the vertical boundary (negative outward), and, along with the fluxes $\Phi_B$ and $\Phi_T$ across the bottom and top of the column, drives the vertical air movement in the column. The only general restrictions on these fluxes are given by (2). To model the situation in the Arctic, we want a downward flow of air with a vertical wind speed, $w$, on the order of 1 mm s$^{-1}$ in the column. A reasonable assumption at the tropopause would be to set $w = 0$, however, since our model has a singularity when $w = 0$, we impose a small wind speed at the tropopause by ensuring $\Phi_T$ is positive. At the





surface boundary layer we impose $\Phi_B \leq 0$. Further, to simplify matters and to ensure a downward flow throughout the column, we assume that

$$\phi(\hat{z}) \leq 0, \qquad \text{if } \hat{z} \in [0, z_c),$$
$$\phi(\hat{z}) \geq 0, \qquad \text{if } \hat{z} \in [z_c, 1],$$

where $z_c$ is some point in $[0, 1]$. The following forms were tested for $\phi(\hat{z})$:

$$\phi(\hat{z}) = \begin{cases} \frac{-1-\Phi_B}{z_c} & \text{if } \hat{z} \in [0, z_c), \\ \frac{1-\Phi_T}{1-z_c} & \text{if } \hat{z} \in [z_c, 1], \end{cases} \qquad \text{piecewise constant,} \qquad (A53)$$

$$\phi(\hat{z}) = \begin{cases} \frac{2(-1-\Phi_B)}{z_c^2}(z_c - \hat{z}) & \text{if } \hat{z} \in [0, z_c), \\ \frac{2(1-\Phi_T)}{(1-z_c)^2}(\hat{z} - z_c) & \text{if } \hat{z} \in [z_c, 1], \end{cases} \qquad \text{piecewise linear,} \qquad (A54)$$

$$\phi(\hat{z}) = \begin{cases} \frac{-1-\Phi_B}{z_c} g_1\left(1 - \frac{\hat{z}}{z_c}, L_{\phi B}\right), & \text{if } \hat{z} \in [0, z_c), \\ \frac{1-\Phi_T}{1-z_c} g_1\left(\frac{\hat{z}-z_c}{1-z_c}, L_{\phi T}\right), & \text{if } \hat{z} \in [z_c, 1], \end{cases} \qquad \text{piecewise sine,} \qquad (A55)$$

$$\phi(\hat{z}) = \begin{cases} \frac{-1-\Phi_B}{z_c} g_2\left(1 - \frac{\hat{z}}{z_c}, L_{\phi B}\right), & \text{if } \hat{z} \in [0, z_c), \\ \frac{1-\Phi_T}{1-z_c} g_2\left(\frac{\hat{z}-z_c}{1-z_c}, L_{\phi T}\right), & \text{if } \hat{z} \in [z_c, 1], \end{cases} \qquad \text{piecewise cosine,} \qquad (A56)$$

where $g_1$ and $g_2$ are defined by (A52). (In the case that $z_c = 0$ or $z_c = 1$, it is understood that only the non-empty interval for
$\phi$ in the above definitions is used and that it is closed at both ends.)

**Appendix B: Model Parameters and Calibration**

This appendix lists the parameter values used in the model and discusses how some of them were calibrated to empirical data. Section B1 gives the calculation of the average annual insolation for the Earth north of 70 N.

Values of most of the model parameters are given in Tables B1 and B2. The parameters in Table B1 are physical constants
except for the last two. The surface horizontal wind speed, $U$, which is a factor in the sensible and latent heat transport from the surface, was set to 10 m/s. The exact value is not too important since $U$ always appears multiplied by the factor $C_D$, which we calibrate to data below. The height of the boundary layer was set to $z_B = 50$ m. The model is not very sensitive to this parameter.

The parameters in Table B2 are those that depend on geographic location. The model was applied to the globally averaged
situation for the purposes of calibration of some parameters (see below) and then also applied to the Arctic. The height of the tropopause is about 9 km at the poles and 17 km at the equator so we used the lower value for the Arctic, and a middle value of 14 km for the global average. The globally averaged insolation, the atmospheric solar reflection, and the average surface albedo are all obtained from Wild et al. (2013). For the Arctic, the insolation is the annual average for the region north of 70 N, the calculation of which is shown in Section B1. For the average global situation there is no ocean or atmospheric heat transport;



**Table B1.** Fixed model parameters

| Parameter | Symbol | Value | Units |
|---|---|---|---|
| reference temperature | $T_R$ | 273.15 | K |
| Stefan-Boltzmann constant | $\sigma$ | $5.67037 \times 10^{-8}$ | W m$^{-2}$K$^{-4}$ |
| latent heat of vaporization for water | $L_v$ | $2.2558 \times 10^6$ | m$^2$s$^{-2}$ |
| specific heat capacity of dry air at $T_R$ | $c_v$ | 716.4 | J K$^{-1}$ kg$^{-1}$ |
| saturated vapour density at $T_R$ | $\rho_w^{sat}(T_R)$ | $4.849 \times 10^{-3}$ | kg m$^{-3}$ |
| universal gas constant | $R$ | 8.31446 | J mol$^{-1}$ K$^{-1}$ |
| molecular weight of $CO_2$ | $M_{CO_2}$ | $4.4009 \times 10^{-2}$ | kg mol$^{-1}$ |
| molecular weight of dry air | $M_A$ | $2.89644 \times 10^{-2}$ | kg mol$^{-1}$ |
| molecular weight of water | $M_W$ | $1.80153 \times 10^{-2}$ | kg mol$^{-1}$ |
| gas constant for air | $R_A = R/M_A$ | 287.058 | m$^2$ s$^{-2}$ K$^{-1}$ |
| specific gas constant of water vapour | $R_W = R/M_W$ | 461.4 | m$^2$ s$^{-2}$ K$^{-1}$ |
| pressure at surface | $P_0$ | 101,325 | Pa |
| standard dry density at $T_R$ | $\rho_0 = P_0/(R_A T_R)$ | 1.29225 | kg m$^{-3}$ |
| conductivity of air | $k$ | $24.35 \times 10^{-3}$ | W m$^{-1}$ K$^{-1}$ |
| gravitational acceleration | $g$ | 9.8 | m s$^{-2}$ |
| surface horizontal wind speed | $U$ | 10 | m s$^{-1}$ |
| height of boundary layer | $z_B$ | 50 | m |

the corresponding values for the Arctic come from Mayer et al. (2019). The Arctic atmospheric reflection and surface albedo also come from Mayer et al. (2019). Relative humidity is low at the top of the troposphere so in both the global and Arctic cases was set to 10%. The surface relative humidity was set to 75% for the global average and 70% for the Arctic. For the global model it is appropriate to assume that the average vertical wind speed is zero, but since the model requires a nonzero wind speed we set $M_{\max} = 2.0 \times 10^{-6}$ kg m$^{-2}$ s$^{-1}$, and we set $\Phi_B = -1$, $\Phi_T = 0.2$, and assumed $\phi(\hat{z})$ is given by (A55)

with $z_c = 0$ and $L_{\phi T} = 1$. ($L_{\phi B}$ is irrelevant since $z_c = 0$.) These settings make the wind speed relatively constant and on the order of $10^{-3}$ mm/s, which is far enough away from the singularity to avoid convergence issues, but is small enough so all of the convection-related terms in the model become negligible. Mass flux for the Arctic situation was set with trial and error to $8.0 \times 10^{-4}$ kg m$^{-2}$ s$^{-1}$, which gave vertical wind speeds in the column on the order of 0.5 mm/s. Since $F_A^{\mathrm{tot}} = 0$ for the global case, the form of $\psi$ and therefore also the parameter $L_\psi$ are not relevant.

Calibration of the other model parameters was done in two steps. First, the absorption coefficients, $k_S$, $k_C$, $k_{W2}$, and $k_{Cl}$, the decay for sensible and latent heat transport, $b$, and the drag coefficient, $C_D$, (which is a multiplicative factor of both $B_2$ and $B_3$) were calibrated using global average energy fluxes obtained from Wild et al. (2013). In addition, this calibration attempted to match estimates from Schmidt et al. (2010) indicating that 25% of absorption is due to carbon dioxide, 25% due to clouds




**Table B2.** Model parameters dependent on geographic location

| Parameter | Symbol | Global Value | Arctic Value | Units |
|---|---|---|---|---|
| top of troposphere | $z_T$ | 14,000 | 9,000 | m |
| insolation | $Q$ | 340 | 185 | W m$^{-2}$ |
| atmospheric solar reflection | $Q_R$ | 76 | 20 | W m$^{-2}$ |
| surface albedo | $\alpha^{\dagger}$ | 24/185 | 2/3 | — |
| ocean heat transport | $F_O$ | 0 | 15 | W m$^{-2}$ |
| total atmospheric heat transport | $F_A^{\text{tot}}$ | 0 | 100 | W m$^{-2}$ |
| relative humidity at tropopause | $\delta_T$ | 0.1 | 0.1 | — |
| relative humidity at bottom | $\delta_B$ | 0.75 | 0.7 | — |
| mass flux | $M_{\text{max}}$ | $2.0 \times 10^{-6}$ | $8.0 \times 10^{-4}$ | kg m$^{-2}$ s$^{-1}$ |
| relative mass flux through tropopause | $\Phi_T$ | 0.2 | 0.05 | — |
| relative mass flux through bottom | $\Phi_B$ | -1 | fitted | — |
| relative zero location of $\phi$ | $z_c$ | 0 | fitted | — |
| $\phi$ length scale top | $L_{\phi T}$ | 1 | fitted | — |
| $\phi$ length scale bottom | $L_{\phi B}$ | — | fitted | — |
| $\psi$ length scale | $L_{\psi}$ | — | fitted | — |
| CO$_2$ level | $\mu$ | 390* | 390* | molar ppm |

$^{\dagger}$For the purposes of model calibration, both $\alpha_c$ and $\alpha_w$ were set to the same value.

*Values used for calibration only.

and 50% due to water vapour. These fractions are determined from the model via

$$Tot = \int_0^1 \hat{\kappa}(y_2, y_7) y_4 \, d\hat{z}, \qquad\qquad C_{\text{contrib}} = \frac{1}{Tot} \int_0^1 G_C \mu y_2 y_4 \, d\hat{z},$$

$$Cl_{\text{contrib}} = \frac{1}{Tot} \int_0^1 G_{Cl} y_4 \, d\hat{z}, \qquad\qquad W_{\text{contrib}} = \frac{1}{Tot} \int_0^1 \frac{G_{W2}\delta}{y_7} e^{G_{W1}(1-1/y_7)} y_4 \, d\hat{z}.$$

The relevant data are given in Table B3.

Using these parameter settings we minimized the sum of squares of the differences between the data from Table B3 (after nondimensionalization) with the model outputs allowing the parameters $k_S$, $k_C$, $k_{W2}$, $k_{Cl}$, $b$, and $C_D$ to vary. In the minimization calculation, the terms associated with the contribution values (last three columns of Table B3) were given a heuristic weight of 0.01, since these values are less reliable than the other data. The resulting calibrated values for the parameters are given in Table B4; the results of the fitting are given in Table B3 and Figure B1. As can be seen in Table B3, the minimization achieved very good agreement with the globally averaged data.



**Table B3.** Global Average Energy Fluxes (W m$^{-2}$) from Wild et al. (2013) and contribution fractions for absorption from Schmidt et al. (2010).

|  | $I_+(z_T)$ | $I_+(0)$ | $I_-(0)$ | $I_S(0)$ | $F_C(0)$ | $C_{\text{contrib}}$ | $Cl_{\text{contrib}}$ | $W_{\text{contrib}}$ |
|---|---|---|---|---|---|---|---|---|
| Data | 239 | 397 | 342 | 185 | 105 | 0.25 | 0.25 | 0.50 |
| Model | 239.7 | 397.4 | 341.7 | 184.9 | 105.2 | 0.2332 | 0.2130 | 0.5538 |

**Table B4.** Calibrated absorption parameter values.

| $k_S$ | $k_C$ | $k_{W2}$ | $k_{Cl}$ | $b$ | $C_D$ |
|---|---|---|---|---|---|
| $4.035 \times 10^{-5}$ | 0.1552 | 0.04969 | $7.020 \times 10^{-5}$ | $4.153 \times 10^{-4}$ | $3.180 \times 10^{-3}$ |

Using the calibrated values for $k_S$, $k_C$, $k_{W2}$, $k_{Cl}$, $b$, and $C_D$ obtained from the first step, the second calibration step was to

595 select the parameters for the functions $\phi(\hat{z})$ and $\psi(\hat{z})$ to attempt to match the annual temperature profile for the Arctic from Cronin and Jansen (2016, Fig. 1). Data from that figure of their paper is reproduced in Table B5. The Arctic values of the geographic-dependent parameters from Table B2 were used. Using each of the four forms for $\phi$, given by Eqns. (A53)–(A56), and each of the two forms for $F_A$, given by Eqns. (A50)–(A51), the sum of the square of the differences between the model and the data in Table B5 was minimized by allowing the parameters $z_c$, $\Phi_B$, $L_{\phi T}$, $L_{\phi B}$, and $L_\psi$ to vary. The fit quality is shown in

Figure B2. From this figure it is evident that the first two forms for $\phi$, namely equations (A53) and (A54) do not give adequate fits. The other two forms for $\phi$ are similar, and both forms of $\psi$ only give small changes. The best fit is the third form for $\phi$ and the second form for $\psi$, that is, equations (A55) and (A51) and so these forms were chosen for the model. The calibrated parameters for these forms of $\phi$ and $\psi$ are given in Table B6 and the corresponding functions $\phi$ and $\psi$ are shown in Figure B3.

For the above calibrations, the albedo values $\alpha_c$ and $\alpha_w$ were treated as the same constant in order to ensure the empirical

albedo value was matched regardless of the surface temperature. Now that the calibrations are complete, we wish to apply the

**Table B5.** Annual Arctic temperature data from Figure 1 of Cronin and Jansen (2016).

| Pressure (kPa) | 100 | 95 | 90 | 85 | 80 | 75 | 70 | 65 | 60 | 55 |
|---|---|---|---|---|---|---|---|---|---|---|
| Temperature (K) | 260.1 | 261.8 | 262.1 | 261.3 | 260.1 | 258.1 | 255.7 | 252.7 | 249.5 | 245.4 |

| Pressure (kPa) | 50 | 45 | 40 | 35 | 30 | 25 | 20 | 15 | 10 | |
|---|---|---|---|---|---|---|---|---|---|---|
| Temperature (K) | 241.4 | 236.3 | 231.4 | 225.7 | 221.3 | 220.1 | 221.8 | 222.5 | 221.5 | |



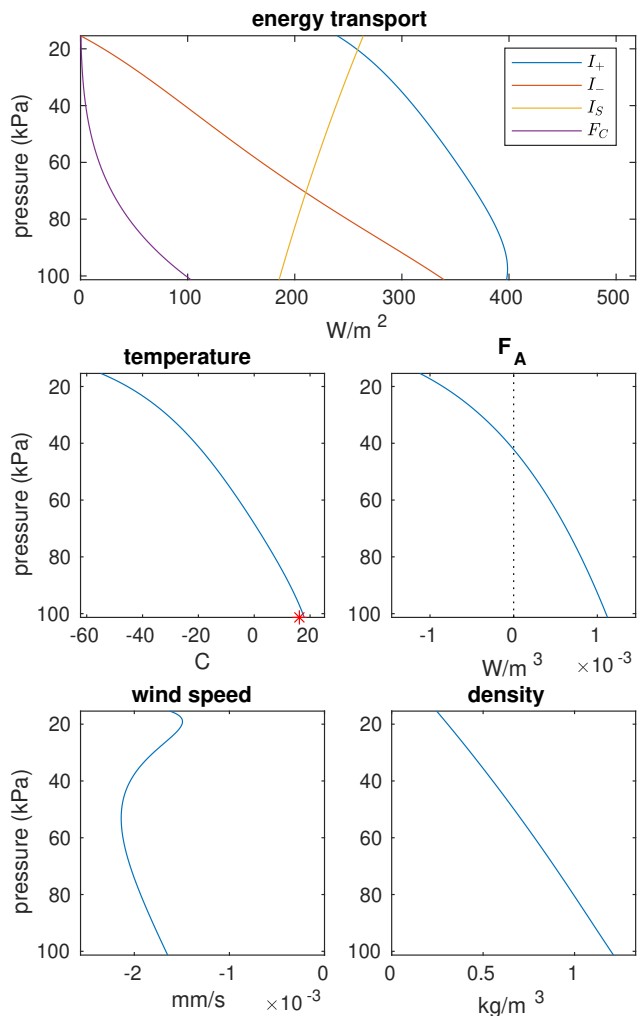

**Figure B1.** Model calibrated to global average values The red asterisk in the middle left plot marks $T_S$.

**Table B6.** Calibrated parameter values for $\phi$ given by (A55) and $\psi$ given by (A51).

| $z_c$ | $\Phi_B$ | $L_{\phi B}$ | $L_{\phi T}$ | $L_\psi$ |
|-------|----------|--------------|--------------|----------|
| 0.2708 | -0.4287 | 1.000 | 0.5727 | 0.7744 |


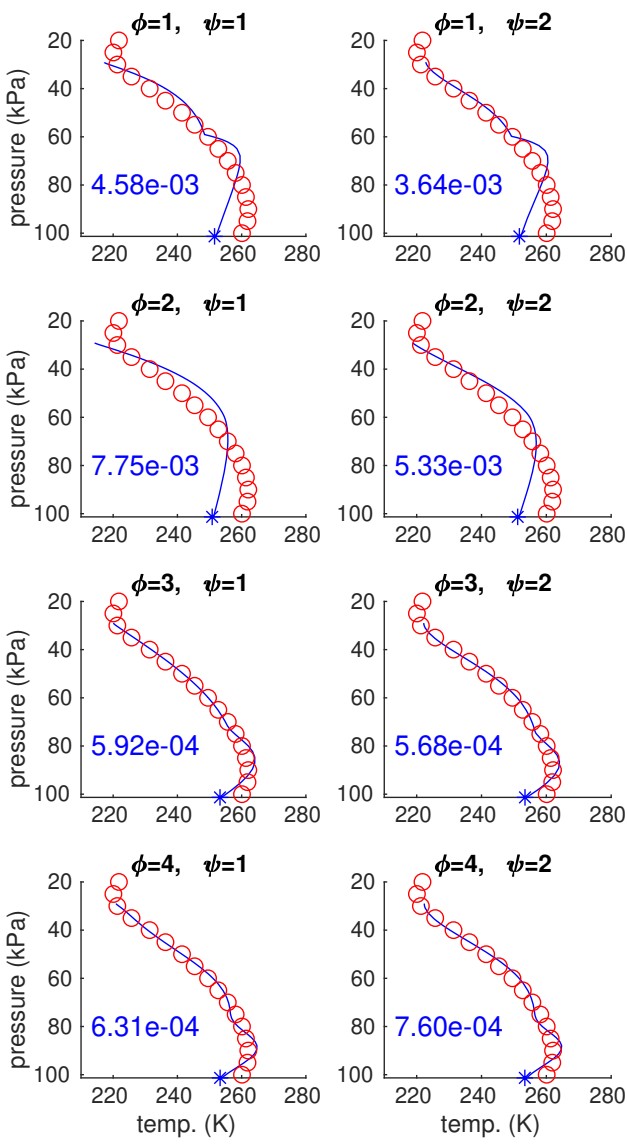

**Figure B2.** Temperature profiles for best fits for each of the $\phi$ (rows) and $\psi$ (columns) forms. The subfigure labels $\phi = m$ for $m = 1, 2, 3, 4$, correspond to Eqns. (A53)–(A56), respectively, and the labels $\psi = n$ for $n = 1, 2$, correspond to Eqns. (A50)–(A51), respectively. The data from Cronin and Jansen (2016) are the red circles. The numbers in the bottom left are the residuals for the fits.

model to the Arctic allowing for the albedo to change with surface temperature. The values of $\alpha_c$, $\alpha_w$, and $\omega$ were chosen as follows. The empirical albedo of 2/3 was used in the Arctic calibration, which resulted in a surface temperature of $T_S = 253.4$ K. Since this temperature is well below freezing, the albedo at this temperature should be near the maximum albedo, so $\alpha_c$ was set to 0.667. Second, $\alpha_w$ was set to 0.1, corresponding to the fact that north of 70 latitude, the Earth is mostly ocean

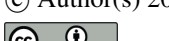
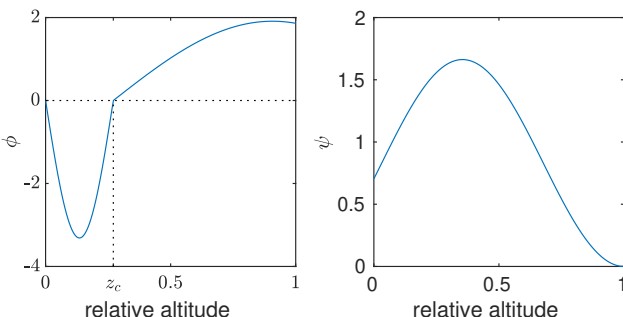

**Figure B3.** Calibrated functions $\phi$ and $\psi$.

**Table B7.** Albedo values for Arctic model, post calibration.

| $\alpha_c$ | $\alpha_w$ | $\omega$ |
|---|---|---|
| 0.667 | 0.1 | 0.01942 |

covered (albedo 0.06 for open water) and partly land (albedo 0.1–0.4). The value of $\omega$ was then calculated from (16) so that $\alpha(253.4) = 2/3$. This resulted in $\omega = 0.01942$. These values are summarized in Table B7.

## B1   Insolation

This section presents the calculation of the insolation used in the Arctic model, where, in particular, the insolation is taken as an annual average over the region north of 70 N.

Select a Cartesian coordinate system $(x, y, z)$ for the solar system with the sun at the origin, with the $z$-axis perpendicular to the Earth's orbital plane, and with the positive $x$-axis defined by the direction in the orbital plane from the sun to the centre of the Earth when the Earth's north pole is furthest from the sun (northern hemisphere winter solstice). Let $(r, \theta)$ be the usual polar coordinates for the centre of the Earth on the orbital plane. Approximate the incoming solar radiation to the Earth as parallel rays traveling from the direction $q = -[\cos\theta, \sin\theta, 0]^T$ with energy flux $S_0 = 1366$ W m$-2$ (the solar constant). Let

$\phi$ and $\psi$ be the latitude and "longitude" of a location on the Earth's surface, where we assume that $\psi = 0$ is aligned with the positive $x$-axis of the solar coordinate system, not with some fixed location on the Earth's surface. If the Earth's axis of rotation was parallel to the $z$-axis (no tilt) then the unit outward normal to the Earth's surface in the solar coordinate system would be

$$\begin{bmatrix} \cos\phi\cos\psi \\ \cos\phi\sin\psi \\ \sin\phi \end{bmatrix}, \qquad -\frac{\pi}{2} \leq \phi \leq \frac{\pi}{2},\ -\pi < \phi \leq \pi.$$





However, the Earth's axis is actually tilted by an angle $\beta = \frac{23.5}{180}\pi$ in the negative sense in the $(x,z)$-plane. Applying this tilt gives the unit outward normal as

$$
n = \begin{bmatrix} \cos\beta & 0 & \sin\beta \\ 0 & 1 & 0 \\ -\sin\beta & 0 & \cos\beta \end{bmatrix} \begin{bmatrix} \cos\phi\cos\psi \\ \cos\phi\sin\psi \\ \sin\phi \end{bmatrix} = \begin{bmatrix} \cos\beta\cos\phi\cos\psi + \sin\beta\sin\phi \\ \cos\phi\sin\psi \\ -\sin\beta\cos\phi\cos\psi + \cos\beta\sin\phi \end{bmatrix}.
$$

The insolation striking the Earth's surface at $(\phi, \psi)$ is then $S_0 \max(q \cdot n, 0)$, where the maximum is due to the fact that the dot product is negative for points on the dark side of the Earth, away from the sun, and hence the insolation there is zero.

Let $S$ be the region of the Earth between latitudes $\phi_1$ and $\phi_2$. The area element is $dS = R^2 \cos\phi \, d\psi \, d\phi$, where $R$ is the radius of the Earth. Therefore the average annual ($\theta$ runs 0 to $2\pi$) insolation on a region, $S$, of the Earth is

$$
Q = \frac{\int_0^{2\pi} \int_S S_0 \max(q \cdot n, 0) \, dS \, d\theta}{\int_S dS} = \frac{S_0}{2\pi(\sin(\phi_2) - \sin(\phi_1))} \int\limits_0^{2\pi} \int\limits_{\phi_1}^{\phi_2} \int\limits_{-\pi}^{\pi} \max(q \cdot n, 0) \cos\phi \, d\psi \, d\phi \, d\theta.
$$

Numerical integration of the above formula yields $Q = 341$ W m$^{-2}$ for the entire globe, which agrees well with Wild's value of 340 (the difference is likely due to some ambiguity in the precise value of $S_0$), and yields $Q = 185$ W m$^{-2}$ for the Arctic region north of 70° latitude.

*Author contributions.* KK roles: formal analysis, investigation, methodology, software, validation, visualization, writing — review & editing. WL roles: conceptualization, funding acquisition, writing — review & editing. GL roles: methodology, software (supporting), funding acquisition, writing — review & editing. AW roles: methodology, writing — original draft, software (supporting), funding acquisition, visualization, writing — review & editing.

*Competing interests.* The authors have no competing interests.

*Acknowledgements.* We acknowledge the support of the Natural Sciences and Engineering Research Council of Canada (NSERC). KK acknowleges the Ontario Ministry of Colleges and Universities and the University of Guelph for a Queen Elizabeth II Graduate Scholarship in Science and Technology.





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
