# Peer review of "Climate Bifurcations in a Schwarzschild Equation Model of the Arctic Atmosphere"

_Nonlinear Processes in Geophysics, 2022_

## Author Response (AR1)

Dear Editor:

Below is a response to the reviewers' comments and a detailed description of alterations made to the manuscript. Page and line numbers of edits mentioned below refer to the **diff.pdf** file.

**Reviewer 1**:

We agree with the first reviewer that there are many details in the paper. They are included because we feel it important that the model be fully explained and so that the results could be replicated. We attempted to strike an appropriate balance between material in the text itself, and what should be put in the appendices. We feel strongly that the appendix material is not simply an addendum, but is essential to the paper, which concurs with the second reviewer's request that we promote reading of the appendices. The first reviewer expressed a desire that the appendices be more readable, but at the same time, the second reviewer commended the detailed explanation.

Changes made:

We made a few more references within the paper itself, (page 4 lines 85-87; page 5 lines 129-1.3) directing the reader to the appendices for further information, and we edited the appendices to help enhance the clarity of the main points from each subsection therein (page 20 lines 473-478; page 22 lines 517-521; page 23 lines 532-534, 541.342, 548-554). In particular we edited Appendix B1 substantially so that all of the parameter values are presented in two tables rather than spread out over five. We also made minor wording changes throughout the main text to help enhance clarity (absract lines 7 and 9; page 9 lines 215-217; page 9 lines 232-234; page 12 lines 301-302 and 308; page 16 line 421).

**Reviewer 2**:

The main concerns of the second reviewer were 1) that we were not sufficiently clear about the aspects of the Arctic climate that do not fit our assumptions, particularly the Arctic Ocean, and 2) that our model is not a simplification of a "more complete" model. The first concern is a valid criticism, and we edited the paper to emphasize the zonal symmetry being assumed and how this symmetry is not present in the Arctic Ocean. However, we do not agree with the reviewer's second assertion. Zonal symmetry of the Earth's atmosphere is a reasonable and well-utilized approximation for a simple annually averaged model of climate. Our current model with a cylindrical atmosphere can be thought of as taking a limit as one restricts the Earth's atmosphere to a vanishingly small region centered at the North Pole. In this limit, the zonally averaged atmosphere becomes a one-dimensional column with downward flow and the PDEs governing the atmospheric fluid flow become ODEs. Alternatively, one can view the model as a meridional and zonal average over a cylinder centred at the North Pole. Although the authors had this view of the model from the beginning, and it is expressed in the sixth paragraph of the introduction, we recognize that the manuscript may not adequately convey this point of view, hence we edited to emphasize how our model arises from this simplification/limiting process. Further, once the model is understood as representing a small region around the North Pole or an appropriately averaged column, the use of a single scalar to represent ocean heat transport becomes a reasonable approximation, regardless of the fact that the ocean is not zonally symmetric. For calibration of our model we used values of ocean and atmospheric heat transport ($F_O$ and $F_A$) measured at 70N latitude. Although this clearly does not correspond to a small region around the North Pole, measured values of these heat transports are not readily available further north. Further, and partly to alleviate the concern around what values of $F_O$ and $F_A$ are used, we did a bifurcation analysis varying these two parameters, showing our conclusions are generic regardless of the precise values used (old Figure 5). We also used the solar insolation value, Q, for the portion of the Earth north of 70N latitude. This value came to 1.3 W m$^{-2}$. This value does not change much by restricting to a region closer to the pole; the limiting value is 173.8 W m$^{-2}$.

Changes made:

We edited the abstract to say "Arctic atmosphere" rather than "Arctic atmosphere-ocean system" to emphasize that the model is primarily an atmospheric mode. A similar change was made on page 3 line 59. An additional paragraph was added on page 5 lines 112-122 discussing zonal symmetry and how our model arising from a limiting process. On page 32, lines 694-695 we mentioned the value of the solar insolation in the case where the column is shrunk to the North Pole.

Responses to detailed comments of the second reviewer:

1) In the past, 1D column/box models have been used to describe the globally-averaged climate of the Earth. These provide little detail and generally arise from the gross approximation of the entire atmosphere as a uniform slab or

altitude-varying column. Here instead, our 1D model results from the assumption of a zonally symmetric atmosphere, making the polar axis invariant, and the limiting process as one considers a small region centered at the North Pole. Thus we believe our 1D model is relevant there. Furthermore, the Earth's climate is changing most rapidly in the high Arctic, so a polar model can be informative. To our knowledge, a 1D polar model has not been studied before.

Changes made:

See changes to main comment of this reviewer that are described above.

2) It is true that the Arctic Ocean is not zonally symmetric, if one is considering the entire Arctic. However, if one is considering a small region around the North Pole, as explained above, this problem is minimized, and a single number can represent ocean heat transport.

Changes made:

See changes to main comment of this reviewer that are described above.

3) The stratosphere is not part of the model, as was recognized by the reviewer. Since the air density is very small in the stratosphere there will be minimal absorption of radiation in the stratosphere, however the effect is not zero. As part of our modelling efforts, we did investigate a simplified stratosphere model attached to the present model, however the resulting quantitative changes to the radiation terms were considered not sufficiently large to warrant the additional complication of modelling the stratosphere. (Actually, the manuscript contained a notation in equations (A17) and (A24) that was a hold-over from our stratosphere modelling that did not get edited out; these equations refer to the downward longwave radiation at the troposphere being the constant $I_-^{TP}$ and its non-dimensional version $K_-$. These have been removed and replaced with zero.)

Changes made:

page 19, line 462: replaced $K_-$ with 0. page 20, Equation (A24), definition of $K_-$ has been removed.

4) The reviewer requested that we add a schematic figure of the model at its introduction.

Changes made:

New Figure 2 added on page 6 and reference to it on page 5 line 129.

5) The reviewer indicates that the model lacks moisture in the main troposphere. This is a mis-reading by the reviewer. The entire atmosphere has moisture content governed by the Clausius-Clapeyron equation and a linear decay of the relative humidity with altitude. These things are discussed in sections 2.1.3 and A.3.1. The absorption of long-wave radiation due to moisture in the air is the third term in the expression for $\kappa$ given in Equation (8). Perhaps the reviewer's oversight was due to the fact that just the symbol $\kappa$ appears in Equation (11) and onward. Water vapour feedback is certainly essential to our model.

Changes made:

Added a sentence to the abstract indicating that water vapour is incorporated in the model through the Clausius-Clapeyron equation. Also indicated in the abstract that the water vapour and surface albedo responses are positive feedbacks. We added a sentences on page 9 lines 221-225 to emphasize that water vapour and carbon dioxide effects are incorporated in the symbol $\kappa$ and that they are essential feedbacks in our model.

6) We thank the reviewer for the commendation; we put considerable effort into explaining the model sufficiently so that it could be replicated.

7) Requested alterations to Table B2, etc.

Changes made:

We edited to refer to Table B2 earlier in the text (page 5 line 134), and added equation numbers to the table as requested. In addition, to aid readability, we combined the values from (old) Tables B4, B6, and B7 into Table B2. This required alterations of the text in Appendix B also.

8) Reviewer requested some alterations to the figures.

Changes made:

We made adjustments to the figures, including wider lines, panel labels, expanded captions, and legends, as requested.

**Other changes made to the manuscript:**

We replaced the variable notation $M_{max}$ with $M_{tot}$ throughout the manuscript (most changes are in Section 2) since it refers to a total amount, not a maximum.

We replaced several references to $F_{Amax}$ with $F_A^{tot}$, as they should have been.

We mentioned the fact that relative humidity is also constant in the boundary layer, page 9 line 238, and page 11 line 282.

We replaced the constant $F_{A0}$ with $F_{A1}$ as the new name is more consistent with the fact that it is the value of $F_A$ at 1.

Imposed consistent spelling of the word "vapour".

Added a reference we failed to include earlier, page 26 line 621, and references to a database and associated publication that we had accessed, page 27 line 622. The reference list was correspondingly updated, page 34 lines 730-733, page 35 lines 748-750, and page 35 lines 760-761.

Fixed a few other minor typos.